# Novel regulators of islet function identified from genetic variation in mouse islet Ca²⁺ oscillations

**Christopher H Emfinger[1†], Lauren E Clark[1†], Brian Yandell[2], Kathryn L Schueler[1], Shane P Simonett[1], Donnie S Stapleton[1], Kelly A Mitok[1], Matthew J Merrins[3,4], Mark P Keller[1], Alan D Attie[1,3,5]***

[1]Department of Biochemistry, University of Wisconsin-Madison, Madison, United States; [2]Department of Statistics, University of Wisconsin-Madison, Madison, United States; [3]Department of Medicine, Division of Endocrinology, University of Wisconsin-Madison, Madison, United States; [4]William S. Middleton Memorial Veterans Hospital, Madison, United States; [5]Department of Chemistry, University of Wisconsin-Madison, Madison, United States

**\*For correspondence:**
adattie@wisc.edu

†These authors contributed equally to this work

**Competing interest:** The authors declare that no competing interests exist.

**Abstract** Insufficient insulin secretion to meet metabolic demand results in diabetes. The intracellular flux of Ca²⁺ into β-cells triggers insulin release. Since genetics strongly influences variation in islet secretory responses, we surveyed islet Ca²⁺ dynamics in eight genetically diverse mouse strains. We found high strain variation in response to four conditions: (1) 8 mM glucose; (2) 8 mM glucose plus amino acids; (3) 8 mM glucose, amino acids, plus 10 nM glucose-dependent insulinotropic polypeptide (GIP); and (4) 2 mM glucose. These stimuli interrogate β-cell function, α- to β-cell signaling, and incretin responses. We then correlated components of the Ca²⁺ waveforms to islet protein abundances in the same strains used for the Ca²⁺ measurements. To focus on proteins relevant to human islet function, we identified human orthologues of correlated mouse proteins that are proximal to glycemic-associated single-nucleotide polymorphisms in human genome-wide association studies. Several orthologues have previously been shown to regulate insulin secretion (e.g. ABCC8, PCSK1, and GCK), supporting our mouse-to-human integration as a discovery platform. By integrating these data, we nominate novel regulators of islet Ca²⁺ oscillations and insulin secretion with potential relevance for human islet function. We also provide a resource for identifying appropriate mouse strains in which to study these regulators.

## eLife assessment

The authors provide a **fundamental** resource, detailing genetic variation of nutrient-responsive islet calcium regulation in mice through the lens of proteomics. The evidence for the mechanisms identified using this resource is **compelling** and strongly supported by integration with results from genome-wide association studies in humans. The construction of a streamlined and searchable web interface for the data will maximize their accessibility and utilization by the community.

## Introduction

The majority of gene loci responsible for the genetic variation in type 2 diabetes (T2D) susceptibility affect the function of endocrine cells of pancreatic islets, primarily β-cells (*Dimas et al., 2014*; *Wood et al., 2017*). Variation in β-cell mass and function places boundaries on their capacity to respond to acute and chronic demands for insulin, such as those of overnutrition and insulin resistance

(*Dimas et al., 2014*; *Wood et al., 2017*). Therefore, metabolic challenges are useful in genetic screens because they expose phenotypes that would otherwise remain silent.

The large collection of inbred mouse strains provides us with a wide repertoire of genetic and phenotypic diversity, comparable to that of the entire human population (*Clee and Attie, 2007*). Yet, most mouse studies have been confined to a small number of highly inbred strains (*Clee and Attie, 2007*; *Kebede and Attie, 2014*). It is becoming widely appreciated that gene deletions, nutritional interventions, and drug effects vary widely among mouse strains, as they do in humans (*Clee and Attie, 2007*; *Sittig et al., 2016*). Thus, characterization of the basis for this high level of phenotypic variation is a path to gain deeper insight into the pathophysiology and genetics of a wide range of physiological processes.

The pancreatic β-cell is a nutrient sensor. In response to particular nutrient stimuli (e.g. glucose, amino acids), the β-cells generate ATP and close ATP-dependent $K^+$ channels ($K_{ATP}$), resulting in plasma membrane depolarization (*Lewandowski et al., 2020*; *Foster et al., 2022*; *Merrins et al., 2022*). This leads to an oscillatory influx of $Ca^{2+}$ ions, triggering insulin secretion. The process of secreting insulin and re-compartmentalizing $Ca^{2+}$ ions consumes ATP, and the drop in the ATP/ADP ratio reopens $K_{ATP}$ channels, repolarizing the membrane, and closing membrane $Ca^{2+}$ channels. Consequently, oscillations in metabolism, insulin secretion, and $Ca^{2+}$ are intrinsically linked (*Merrins et al., 2022*; *Dahlgren et al., 2005*; *Krippeit-Drews et al., 2000*; *Marinelli et al., 2022b*; *Henquin, 2009*), and the capacity to maintain functional $Ca^{2+}$ handling has been suggested to be critical for islet compensation (*Chen et al., 2016*).

In this study, we utilized the extraordinary genetic and phenotypic diversity represented in the eight founder mouse strains (which we subsequently refer to as 'founders') used to generate the Collaborative Cross (CC) recombinant inbred mouse panel and the Diversity Outbred (DO) stock (*Threadgill et al., 2011*; *Svenson et al., 2012*). These strains capture most of the genetic diversity of all inbred mouse strains (*Threadgill et al., 2011*; *Svenson et al., 2012*). While studies of these mice have provided significant insight into genetic regulators of islet function (*Keller et al., 2019*), determining the appropriate model system for evaluating genes of interest is often difficult, as most deletion models are made in only a small number of strains, primarily C57BL/6J or C57BL/6N.

We explored the diversity of nutrient-evoked islet $Ca^{2+}$ responses across the eight founder mouse strains, uncovering a remarkable diversity of $Ca^{2+}$ oscillations. Our prior proteomics studies showed that the protein abundance from islets of the founder mouse strains is also highly diverse, as is their insulin secretory response to different stimuli (*Mitok et al., 2018*). By correlating the strain and sex variation in protein abundance with the variation in $Ca^{2+}$ oscillations, we identified a small number of islet proteins that are highly correlated with islet $Ca^{2+}$ oscillations. The human orthologues of many of these proteins are encoded by genes with nearby single-nucleotide polymorphisms (SNPs) linked to glycemic traits (e.g. fasting blood glucose, see Table 2 for terms) in genome-wide association studies (GWAS). By integrating these data, we nominate novel regulators of islet $Ca^{2+}$ oscillations and insulin secretion with potential relevance for human islet function. We provide a web-based resource that integrates proteomic and $Ca^{2+}$ data for identifying appropriate mouse strains in which to study these regulators.

## Results
### Genetics exerts a strong influence on islet $Ca^{2+}$ dynamics

Glucose metabolism, β-cell $Ca^{2+}$ flux, and insulin secretion are pulsatile, and have been found to oscillate in both humans and mice (*Merrins et al., 2022*; *Dahlgren et al., 2005*; *Nunemaker et al., 2006b*; *Kennedy et al., 2002*; *Lang et al., 1979*). Because they are interconnected, understanding the factors governing oscillation patterns can inform about the mechanisms that regulate insulin secretion (*Lewandowski et al., 2020*; *Marinelli et al., 2022b*; *Colsoul et al., 2010*; *Corbin et al., 2016*). To explore the influence of genetic background on $Ca^{2+}$ oscillations, we measured $Ca^{2+}$ in islets of the eight CC founder strains, that together harbor as much genetic diversity as humans: A/J, C57BL/6J (B6), 129S1/SvlmJ (129), NOD/ShiLtJ (NOD), NZO/HILtJ (NZO), CAST/EiJ (CAST), PWK/PhJ (PWK), and WSB/EiJ (WSB).

All mice were maintained on a Western-style diet (WD) high in fat and sucrose for 16 weeks prior to isolating their islets for $Ca^{2+}$ imaging with Fura Red, a $Ca^{2+}$-sensitive fluorescent dye (*Figure 1A*).

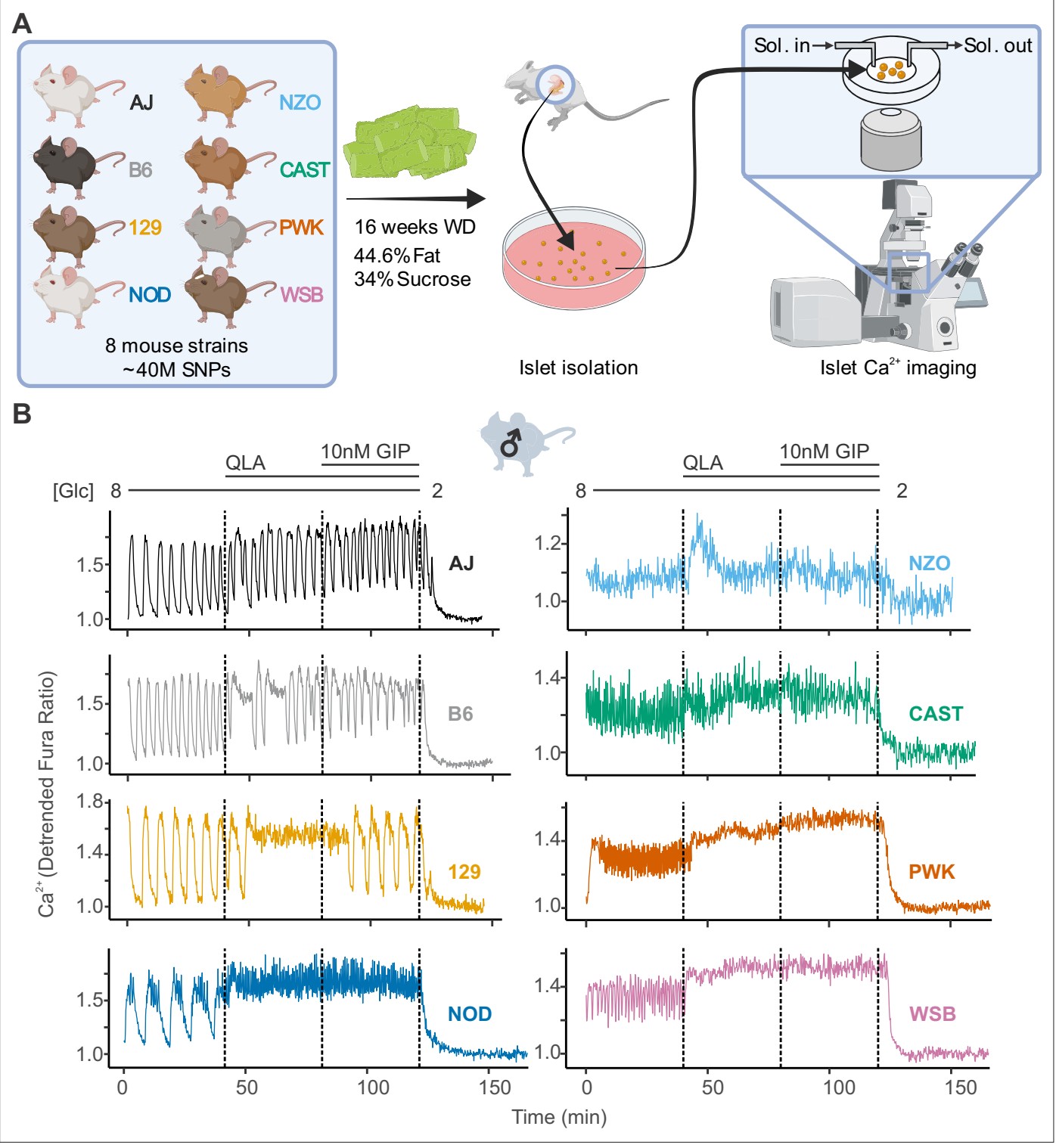

**Figure 1.** High diversity in Ca²⁺ oscillations across eight genetically distinct mouse strains. (**A**) Male and female mice from eight strains (A/J; C57BL/6J (B6); 129S1/SvlmJ (129); NOD/ShiLtJ (NOD); NZO/HlLtJ (NZO); CAST/EiJ (CAST); PWK/PhJ (PWK); and WSB/EiJ (WSB)) were placed on a Western diet (WD) for 16 weeks before their islets were isolated. The islets were then imaged on a confocal microscope using Fura Red dye under conditions of 8 mM glucose (8G); 8G + 2 mM L-glutamine, 0.5 mM L-leucine, and 1.25 mM L-alanine (8G/QLA); 8G/QLA + 10 nM glucose-dependent insulinotropic polypeptide (8G/QLA/GIP); and 2 mM glucose. (**B**) Representative Ca²⁺ traces for male mice (*n* = 3–8 mice per strain, and 15–83 islets per mouse), with the transitions between solution conditions indicated by dashed lines. Abbreviations: '[Glc]' = 'concentration of glucose in mM'; 'Sol.' = 'solution'; 'SNPs' = 'single-nucleotide polymorphisms'.

*Figure 1 continued on next page*

*Figure 1 continued*

The online version of this article includes the following figure supplement(s) for figure 1:

**Figure supplement 1.** The high diversity in $Ca^{2+}$ oscillation in males is also observed in female mice.

**Figure supplement 2.** Diverse responses in non-diabetic NOD females' islets.

We measured $Ca^{2+}$ dynamics in response to four conditions: (1) 8 mM glucose (8G); (2) 8G + 2 mM glutamine, 0.5 mM leucine, and 1.25 mM alanine (8G/QLA); (3) 8G/QLA + 10 nM glucose-dependent insulinotropic polypeptide (8G/QLA/GIP); and (4) 2 mM glucose (2G) (*Figure 1B*). There was a high degree of similarity between three of the five classical strains (A/J, B6, 129), which were dominated by slow oscillations (period 2–10 min) in 8G and 8G/QLA/GIP, and had relatively fewer islets reach plateau (continuous peak activity without oscillation) in 8G/QLA. Likewise, the wild-derived strains (CAST, WSB, and PWK) closely matched one another, while differing from the classical strains. The wild-derived mouse islets were dominated by fast oscillations (period <2 min) in 8G, resulting in plateaus for 8G/QLA and 8G/QLA/GIP.

Two strains stood out from the others. Islets from NOD mice showed characteristics from both the wild-derived and classical strains; slow oscillations in 8G and a sustained plateau in response to 8G/QLA and 8G/QLA/GIP with fast oscillations superimposed. The NZO mice also differed from the other classical strains, likely because they were all diabetic (blood glucose >250 mg/dl). Their islets were minimally responsive to 8G but did respond with a strong pulse in 8G/QLA and $Ca^{2+}$ remained elevated in 8G/QLA/GIP.

Many of the strain differences seen in the male mice were maintained in the females (*Figure 1— figure supplement 1A*). The classic strains were once again highly similar to one another, as were the wild-derived strains. Furthermore, the NZO females, of which all but one were diabetic, mirrored the behavior of the male islets. One interesting observation that emerged from the female islets is that the NOD females displayed a greater variation in their $Ca^{2+}$ oscillations than the NOD males (*Figure 1— figure supplement 2*). Some of the islets maintained slow oscillations throughout the various conditions, while some demonstrated fast oscillations and plateaued like the wild-derived strains. Yet others appeared strikingly similar to the islets from diabetic NZO mice, despite none of the NOD mice being diabetic. Finally, the one non-diabetic female NZO displayed oscillatory behavior comparable to that of the other classical strains, with clear, slow oscillations (*Figure 1—figure supplement 1B*).

## Dissecting islet $Ca^{2+}$ dynamics

An understanding of the mechanisms regulating insulin secretion, including the roles of specific metabolic pathways, ion channels, and hormones, has been derived from the shape and frequency of islet $Ca^{2+}$ oscillations (*Lewandowski et al., 2020*; *Dahlgren et al., 2005*; *Marinelli et al., 2022b*; *Kennedy et al., 2002*; *Nunemaker et al., 2006a*; *Marinelli et al., 2022a*; *Nunemaker et al., 2005*; *Bertram et al., 2018*). To elucidate strain differences in $Ca^{2+}$ dynamics, we focused on six parameters of the $Ca^{2+}$ waveform (*Figure 2A*): (1) peak $Ca^{2+}$ (the maximum value of each oscillation); (2) period (the length of time between two peaks); (3) active duration (the length of time for each $Ca^{2+}$ oscillation measured at half of the peak height, also known the oxidative 'secretory' phase, or 'Mito$_{Ox}$' [*Merrins et al., 2022*]); (4) pulse duration (active duration plus extra time for $Ca^{2+}$ extrusion); (5) silent duration (the electrically silent 'triggering' phase, also known as 'Mito$_{Cat}$' (*Merrins et al., 2022*), which culminates in $K_{ATP}$ closure and membrane depolarization); and (6) plateau fraction (the active duration divided by the period, or the fraction of time spent in the active 'secretory' phase).

We also assessed the spectral density for every islet to extract additional information from complex oscillations where multiple components were visible (*Figure 2—figure supplement 1A*). We analyzed each trace to determine the top two frequencies contributing to the trace (1st and 2nd component frequencies, *Figure 2—figure supplement 1B*) and their respective contributions (1st and 2nd component amplitudes). Because certain features, such as metabolically driven (slow) and electrically driven (fast) oscillations have characteristic frequencies (*Bertram et al., 2007*), extracting the top two frequencies may highlight additional information beyond that previously collected.

A representative $Ca^{2+}$ trace from a female B6 islet is illustrated in *Figure 2A*. The transition from 8G to 8G/QLA resulted in an increased active duration, yielding a longer period and an increased plateau fraction. For an islet that plateaued at the peak, as seen in 8G/QLA (*Figure 2B*), we computed

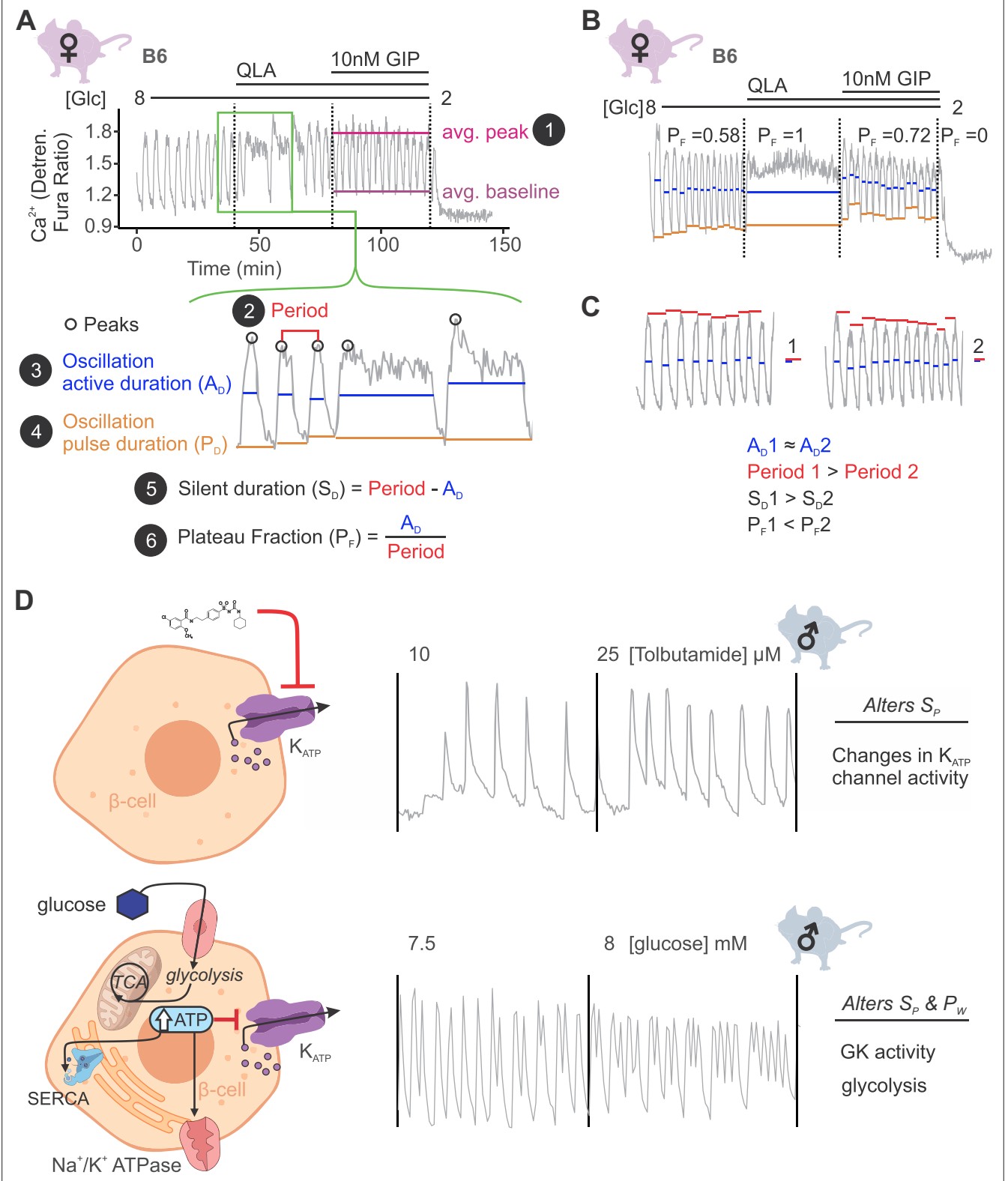

**Figure 2.** Ca$^{2+}$ wave breakdown reveals mechanisms underlying Ca$^{2+}$ responses. (**A**) An example B6 female Ca$^{2+}$ wave, showing that the islet oscillations can change in their average peak (1) and average baseline in response to different nutrients. Additionally, shifts in wave shape (green box) can be broken down into changes in time between peaks (period, 2), the time in the active phase (active duration, $A_D$, 3), and the length of the oscillation (pulse duration, $P_D$, 4). From these, the time inactive between oscillations (silent duration, $S_D$, 5), and the relative time in the active phase, or plateau

*Figure 2 continued on next page*

*Figure 2 continued*

fraction ($P_F$, 6), can be calculated. Each parameter can be changed by different underlying mechanisms. (**B**) For islets that plateaued, as in the example islet in 8G/QLA, they were assigned a plateau fraction of one and a period of zero. For islets that ceased to oscillate, such as the example islet in 2 mM glucose, they were assigned a plateau fraction of zero and a period of the time of measurement (40 min). (**C**) For trace 1 (left), which has a longer period (red bars) than trace 2 (right), but the same active duration (blue bars), the silent duration is greater and consequently the $P_F$ is shorter, in contrast to the trace in (**A**) where the $P_F$ increases between 8G and 8G/QLA are largely due to increases in $A_D$. (**D**) Changes in specific $Ca^{2+}$ wave parameters can reflect different mechanisms in β-cells. For example, changing $K_{ATP}$ activity pharmacologically (upper panels) predominantly increases $P_F$ by altering $S_D$, whereas increasing glucose concentrations by elevating glucose or activating GK cause significant alterations in both $A_D$ and $S_D$ to increase $P_F$. Abbreviations: '[Glc]' = 'concentration of glucose in mM'; 'GK' = 'glucokinase'.

The online version of this article includes the following figure supplement(s) for figure 2:

**Figure supplement 1.** Example of spectral density breakdown for $Ca^{2+}$ traces.

a plateau fraction of one, an active and pulse duration of 40 min (the measurement time), and a period of 0 min. An islet that returned to baseline and ceased to oscillate, as seen in 2 mM glucose (*Figure 2B*), was determined to have a plateau fraction, active duration, and pulse duration of zero, and a period of 40 min. Dissecting the strain dependence of these key parameters of the $Ca^{2+}$ oscillations is important for identifying underlying mechanisms, as illustrated in *Figure 2C*. While both traces have a similar active duration (blue bars), trace 1 has a longer period (red bars), resulting in an increased silent duration and a decreased plateau fraction.

Examples of pathways altering specific components of $Ca^{2+}$ oscillations have previously been established (*Figure 2D*; *Dahlgren et al., 2005*; *Marinelli et al., 2022b*; *Nunemaker et al., 2006b*; *Whitticar and Nunemaker, 2020*; *Koneshamoorthy et al., 2022*). For example, when $K_{ATP}$ channels are pharmacologically closed with tolbutamide, the silent duration is shortened, resulting in increased frequency without a change in pulse shape (upper panel). The addition of glucose leads to increased glucose metabolism and glucokinase (GK) activity (*Marinelli et al., 2022b*). The resulting rise in ATP inhibits $K_{ATP}$ channels (*Lewandowski et al., 2020*) and is used as a substrate for additional processes that affect $Ca^{2+}$, such as SERCA pumps (*Tengholm and Gylfe, 2017*; *Shuai et al., 2021*). Thus, glucose alters both the active and silent durations, resulting in a change in both frequency and shape of the $Ca^{2+}$ oscillations (lower panel).

## Parameters significantly correlated with insulin secretion show remarkable variance by strain and sex

Average $Ca^{2+}$ is commonly used for analyzing $Ca^{2+}$ dynamics and is frequently assumed to be highly correlated to insulin secretion. To determine whether average $Ca^{2+}$ is predictive of insulin secretion, we performed ex vivo perifusion studies on islets from WSB and 129 male mice, two strains that showed similar average $Ca^{2+}$ (*Figure 3—figure supplement 1*) but exhibited vastly different $Ca^{2+}$ oscillations (*Figure 1B*). WSB mice had significantly higher insulin secretion in each of the secretory conditions (*Figure 3A*), suggesting another $Ca^{2+}$ parameter better predicts insulin secretion.

To identify parameters of the $Ca^{2+}$ dynamics most strongly correlated to insulin secretion, we computed the correlation between the $Ca^{2+}$ oscillation parameters and our previously published insulin secretion in similar conditions (8.3G, 8.3G/QLA, basal) for the same sexes and strains (*Figure 4A*, *Figure 4—figure supplement 1*, and *Figure 4—figure supplement 2*; *Mitok et al., 2018*). Consistent with our observations from the perifusion data in the WSB and 129 islets, we found that average $Ca^{2+}$ was not strongly correlated to insulin secretion. Other metrics, such as active duration in 8G, and the silent durations in 8G/QLA, were more highly correlated to insulin secretion. Meanwhile, the 1st component frequency in 8G from the spectral density analysis was highly correlated with *decreased* insulin secretion. These metrics were also the most highly correlated with multiple clinical measures in the founder mice, particularly plasma insulin (*Figure 4B*, *Figure 4—figure supplement 1*, and *Figure 4—figure supplement 3*), for which silent duration in 8G/QLA/GIP had the strongest correlations.

Several parameters of the $Ca^{2+}$ oscillatory waveform showed strong strain and sex effects (*Figure 4C*, **D** and *Figure 4—figure supplement 1*). For example, basal $Ca^{2+}$ (average $Ca^{2+}$ in 2G, *Figure 4C*) was relatively consistent among the strains, except NZO where it was highest in islets from male mice. For the overall pulse duration (*Figure 4D*), the NZO mice were once again the highest, followed by CAST and WSB. A noticeable sex effect was measured for the CAST mice, where male mice had a longer

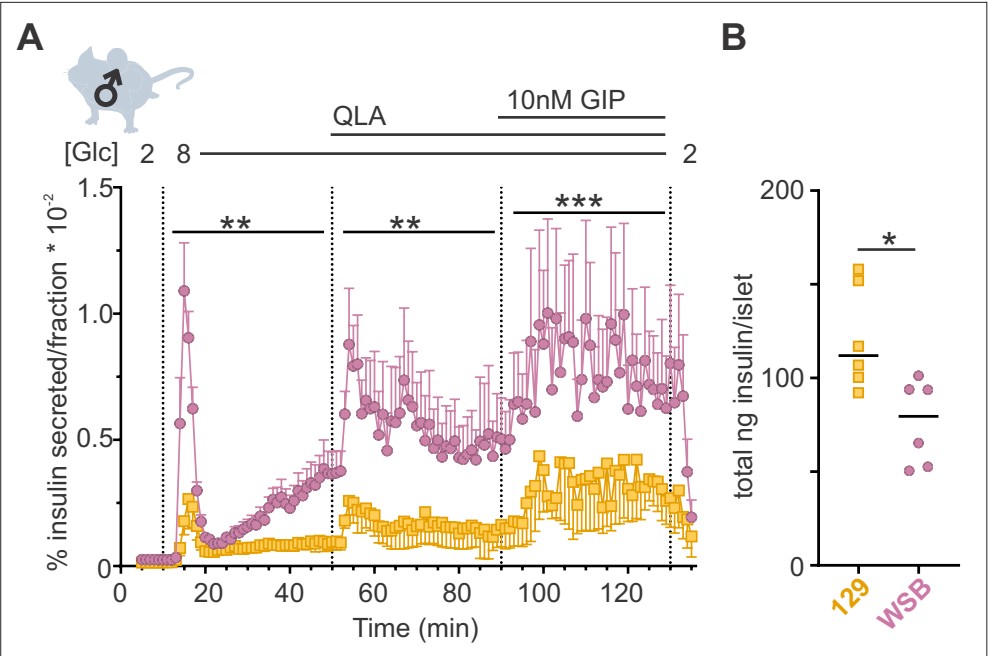

**Figure 3.** WSB mice secrete significantly more insulin than 129 mice. (**A**) Insulin secretion was measured for perifused islets from WSB ($n = 6$, magenta circles) and 129 ($n = 5$, yellow squares) male mice in 2 mM glucose, 8G, 8G/QLA, and 8G/QLA/GIP. Transitions between solutions are indicated by dotted lines and the conditions for each are indicated above the graph. '[Glc]' denotes the concentration of glucose in mM. Data are shown as a percentage of total islet insulin (mean ± standard error of the mean [SEM]). (**B**) Average total insulin per islet for the WSB and 129 males used in (**A**) with one exception: islets from one of the 129 mice were excluded from perifusion analysis due to technical issues with perifusion system on the day those animals' islets were perifused. Dots represent individual values, and the mean is denoted by the black line. For (**A**), asterisks denote strain effect for the area-under-the-curve of the section determined by two-way analysis of variance (ANOVA), mixed effects model; **$p < 0.01$, ***$p < 0.001$. For (**B**), asterisk denotes $p < 0.05$ from Student's $t$-test with Welch's correction.

The online version of this article includes the following figure supplement(s) for figure 3:

**Figure supplement 1.** Average $Ca^{2+}$ for the stimulatory conditions.

pulse duration than the female mice. The 1st component frequency (*Figure 4E*) is driven by the differences observed in the wild-derived strains, for which CAST has the highest frequency, followed by PWK and WSB. Finally, the trend for a sex effect in the classic strains on the silent duration (at 8G, 8G/QLA, and 8G/QLA/GIP) is absent in the NZO and wild-derived mice with the former having greater silent duration in males and the latter frequently having islets plateau in response to these stimuli.

Clustering the $Ca^{2+}$ responses into distinct groups based on our observations of the waveforms (*Figures 1B and 4C–E*, *Figure 1—figure supplement 1*, and *Figure 1—figure supplement 2*) also occurs when correlating individual $Ca^{2+}$ parameters to ex vivo secretion and clinical data (*Figure 4—figure supplement 1*). For example, the anticorrelation between the 1st frequency component in 8G and percent insulin secreted in 8.3G/QLA (*Figure 4—figure supplement 1A*) separates the classical inbred, wild-derived, and diabetes-susceptible strains into distinct groups despite the variability in the trait. Correlation between the silent duration in 8G/QLA to insulin secretion in 8.3G/QLA, likewise groups by strain (*Figure 4—figure supplement 1B*). Finally, some correlations, such as that between 8G/QLA/GIP silent duration and plasma insulin at sacrifice (*Figure 4—figure supplement 1C*), can be strongly influenced by outlier strains; for example NZO. Collectively, these data demonstrate that genetics has a profound influence on key parameters of islet $Ca^{2+}$ oscillations.

## Calcium oscillatory parameters correlate strongly to the abundance of specific islet proteins

To explore relationships between $Ca^{2+}$ oscillations and islet proteins, we took advantage of our whole islet proteomic survey from the eight founder strains (*Mitok et al., 2018*). To identify proteins that may

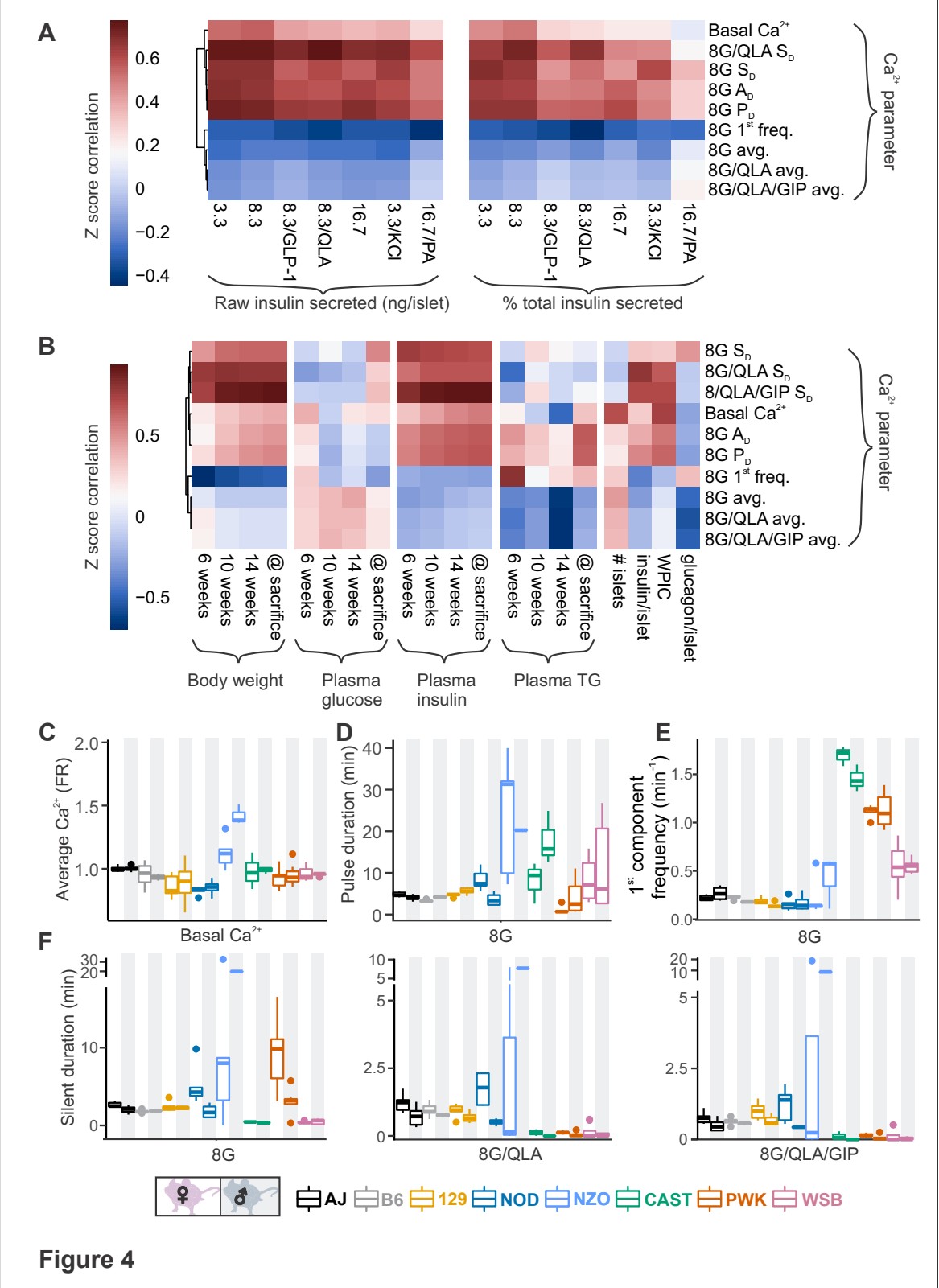

**Figure 4**

**Figure 4.** Comparing sex and strain patterns for Ca²⁺ metrics, insulin secretion, and clinical traits nominates Ca²⁺ metrics of interest. (**A**) The *Z*-score correlation coefficient was calculated for Ca²⁺ parameters and raw insulin secreted and % total insulin secreted. Insulin measurements were previously collected for seven different secretagogues (16.7 mM glucose + 0.5 mM palmitic acid (16.7G/PA); 3.3 mM glucose + 50 mM KCl (3.3G/KCl); 16.7 mM glucose (16.7G); 8.3 mM glucose + 1.25 mM L-alanine, 2 mM L-glutamine, and 0.5 mM L-leucine (8.3G/QLA); 8.3 mM glucose + 100 nM GLP-1 (8.3G/

*Figure 4 continued on next page*

*Figure 4 continued*

GLP-1); 8.3 mM glucose (8.3G); and 3.3 mM glucose (3.3G)) (***Mitok et al., 2018***). (**B**) Correlation of the $Ca^{2+}$ parameters to the clinical measurements in the founder mice which include (1) plasma insulin, triglycerides (TG), and glucose at 6, 10, and 14 weeks as well as at time of sacrifice; (2) number of islets; (3) whole-pancreas insulin content (WPIC); and (5) islet content for insulin and glucagon. For (**A**) and (**B**), the $Ca^{2+}$ parameters shown here include average $Ca^{2+}$ in 2 mM glucose (basal $Ca^{2+}$); average $Ca^{2+}$ in 8 mM glucose (8G avg.); average $Ca^{2+}$ in 8 mM glucose + 1.25 mM L-alanine, 2 mM L-glutamine, and 0.5 mM L-leucine (8G/QLA avg); average $Ca^{2+}$ in 8 mM glucose + QLA + 10 nM GIP (8G/QLA/GIP avg.); pulse duration in 8 mM glucose (8G $P_D$); active duration in 8G (8G $A_D$); silent duration in 8G (8G $S_D$), 8G/QLA (8G/QLA/$S_D$), and 8G/QLA/GIP (8G/QLA/GIP $S_D$); and 1st component frequency in 8 mM glucose (8G 1st freq.). Other parameters analyzed are indicated in ***Figure 4—figure supplement 2*** and ***Figure 4—figure supplement 3***. (**B–E**) Sex and strain variability for (**C**) average $Ca^{2+}$ determined by the Fura-ratio (FR) in 2 mM glucose, (**D**) pulse duration of oscillations in 8G, (**E**) 1st component frequency in 8G, and (**F**) silent duration of oscillations in 8G, 8G/QLA, and 8G/QLA/GIP.

The online version of this article includes the following figure supplement(s) for figure 4:

**Figure supplement 1.** Differential strain and sex effects in correlations between traits.

**Figure supplement 2.** Correlation reveals specific $Ca^{2+}$ parameters highly associated with insulin secretion.

**Figure supplement 3.** Correlation reveals specific $Ca^{2+}$ parameters highly associated with in vivo traits.

underly the strain differences in $Ca^{2+}$ oscillations, we computed the correlation between islet protein abundance and $Ca^{2+}$ dynamics across all mice used in our study (***Figure 5*** and ***Figure 5—figure supplement 1***). Our previous survey of islet proteomics included both sexes for all strains, except NZO males, resulting in a quantitative measure of 4054 proteins (***Mitok et al., 2018***). ***Figure 5A*** illustrates a heatmap of the correlation between islet proteins and several parameters of $Ca^{2+}$ oscillations. Unsupervised clustering was used to show that groups of proteins showed strong positive or negative correlation to a given $Ca^{2+}$ parameter, yielding distinct correlation architecture. For example, proteins highly correlated to the 8G 1st component frequency tended to also be strongly anticorrelated to the silent duration conditions, which were very similar to one another. The active and pulse durations for 8G had nearly identical correlation structure. Additionally, the conditions with the fewest highly correlated proteins were the average $Ca^{2+}$ measures for 8G, 8G/QLA, and 8G/QLA/GIP, and the structure for these was largely inverted from the active duration conditions. Finally, despite the differences in the overall correlations between the different metrics, there were proteins that did overlap (e.g. the block of proteins with high correlation to both 8G $A_D$ and 8G/QLA $S_D$) suggesting that while there were clusters of distinct proteins/pathways for any given metric some proteins may modify more than one metric.

Among the 4054 islet proteins, 363 had high absolute correlation coefficients ($r > |0.5|$) to 3 or more of the parameters our data suggest most strongly correlate to insulin secretion and plasma insulin (Basal $Ca^{2+}$, 8G $A_D$, 8G $P_D$, 8G/QLA $S_D$, 8G $S_D$, 8G/QLA/GIP $S_D$). Interestingly, of the proteins correlated to these traits, many have been previously implicated in islet biology, including PCSK1, GCK, SUR1, GLUT2, PDX1, and GLP-1 (***Whitticar and Nunemaker, 2020***; ***Koneshamoorthy et al., 2022***; ***Tengholm and Gylfe, 2017***; ***Shuai et al., 2021***; ***Koster et al., 2000***; ***Remedi and Nichols, 2016***; ***Jennings et al., 2020***; ***Rutter et al., 2020***; ***Stijnen et al., 2016***). Notably, the highly correlated proteins enriched for tissues, pathways, and transcription factors that support their role in insulin secretion (***Figure 5B–D***, Enrichr links in ***Supplementary file 3***; ***Chen et al., 2013***). For instance, proteins highly anticorrelated to active duration in 8G were enriched for components of oxidative metabolism and had their gene promoters enrich for binding to the islet transcription factor MAFA. These enrichment data provide a framework for discovering new genes of interest for their role in islet function.

## Integration of mouse genetics with human GWAS

The data presented in ***Figure 5A*** illustrate the *correlation* between islet proteins and $Ca^{2+}$ dynamics. Importantly, a protein strongly correlated to $Ca^{2+}$ does not necessarily reflect a causal relationship, that is a change in protein abundance may or may not cause a change in the $Ca^{2+}$ signal. To take our analysis beyond correlation, we integrated our data with human GWAS of glycemia-related traits.

For each $Ca^{2+}$ parameter, we focused on those proteins in the tails of the correlation histogram where $r > |0.5|$ (e.g. ***Figure 5B***). We identified human homologues for 3073 proteins that were correlated to $Ca^{2+}$ in either direction for at least one of our parameters of interest. We then searched the Type 2 Diabetes Knowledge Portal (https://t2d.hugeamp.org/) for SNPs that are associated with one or more glycemia-related traits (see Table 2) with a p-value $<10^{-8}$, and are located within ±100 kbp of the homologous gene (e.g. COBLL1, ***Figure 6A***), or in regions that contact the gene's promoter

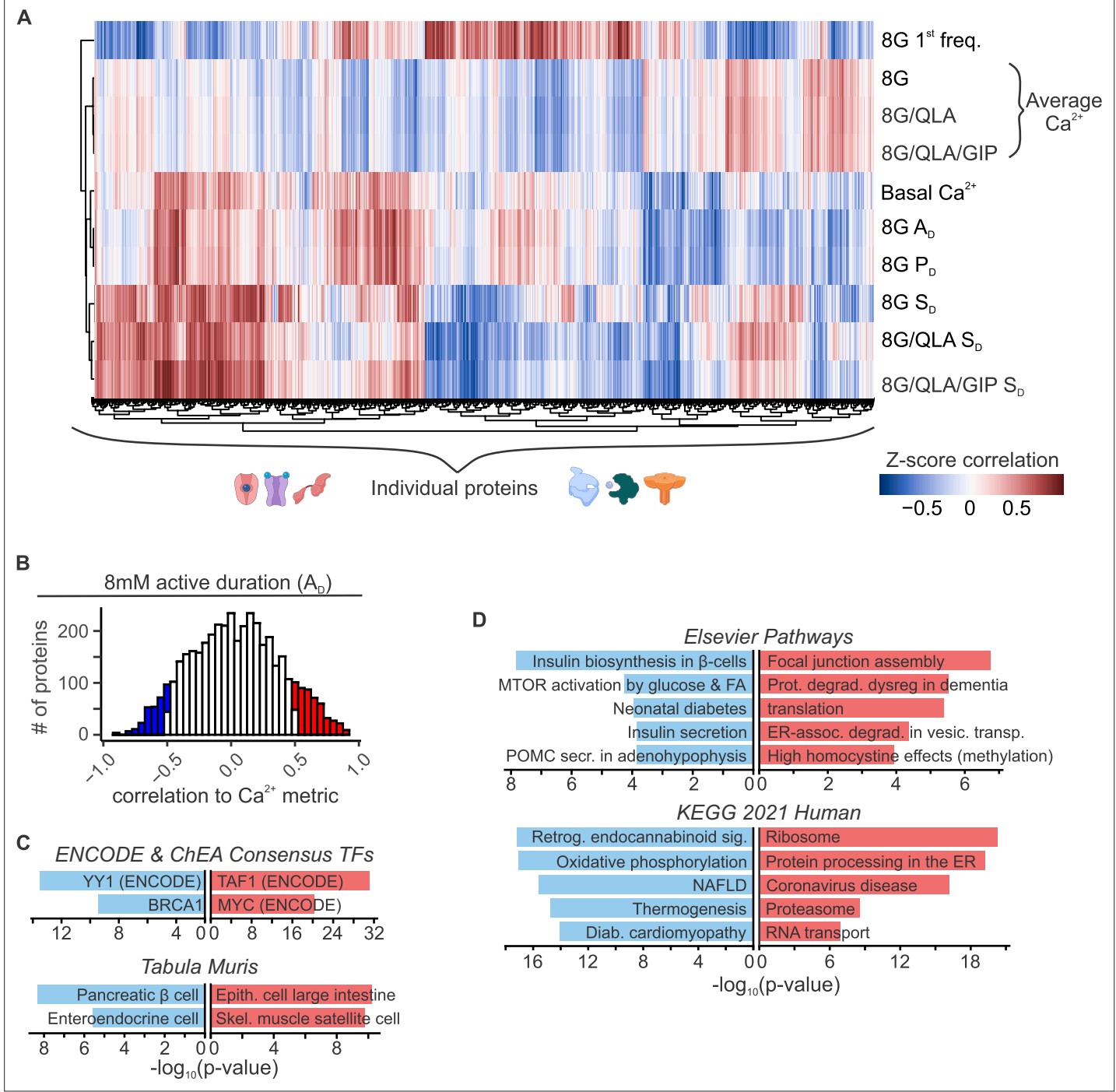

**Figure 5.** Islet proteins show correlation architecture to specific Ca²⁺ parameters. (**A**) Unsupervised clustering of correlation coefficients between protein abundance Z-scores and Z-scores for the Ca²⁺ parameters indicated. Islet proteins show differential correlation values to basal Ca²⁺, excitatory Ca²⁺ (detrended average values for 8G, 8/QLA, and 8/QLA/GIP), active duration and pulse duration in 8G (8G $P_D$ and $A_D$), and silent durations ($S_D$) in 8G, 8G/QLA, and 8G/QLA/GIP. Correlation coefficients for other parameters are indicated in *Figure 5—figure supplement 1*. (**B**) Histograms representing the number of proteins that are correlated (red) and anticorrelated (blue) to 8G $A_D$. ENCODE & CHEA Consensus transcription factor motif database and *Tabula Muris* tissue single-cell RNA-seq signature database (**C**) as well as pathway enrichments for the Elsevier Pathway database and KEGG 2021 Human pathway database (**D**) (−log₁₀(p-values)), for the highly correlated (red) and anticorrelated (blue) proteins to 8 $A_D$ metric. Databases were queried using Enrichr (*Chen et al., 2013*; *Kuleshov et al., 2016*).

The online version of this article includes the following figure supplement(s) for figure 5:

**Figure supplement 1.** Correlation reveals proteins highly associated with specific Ca²⁺ parameters.

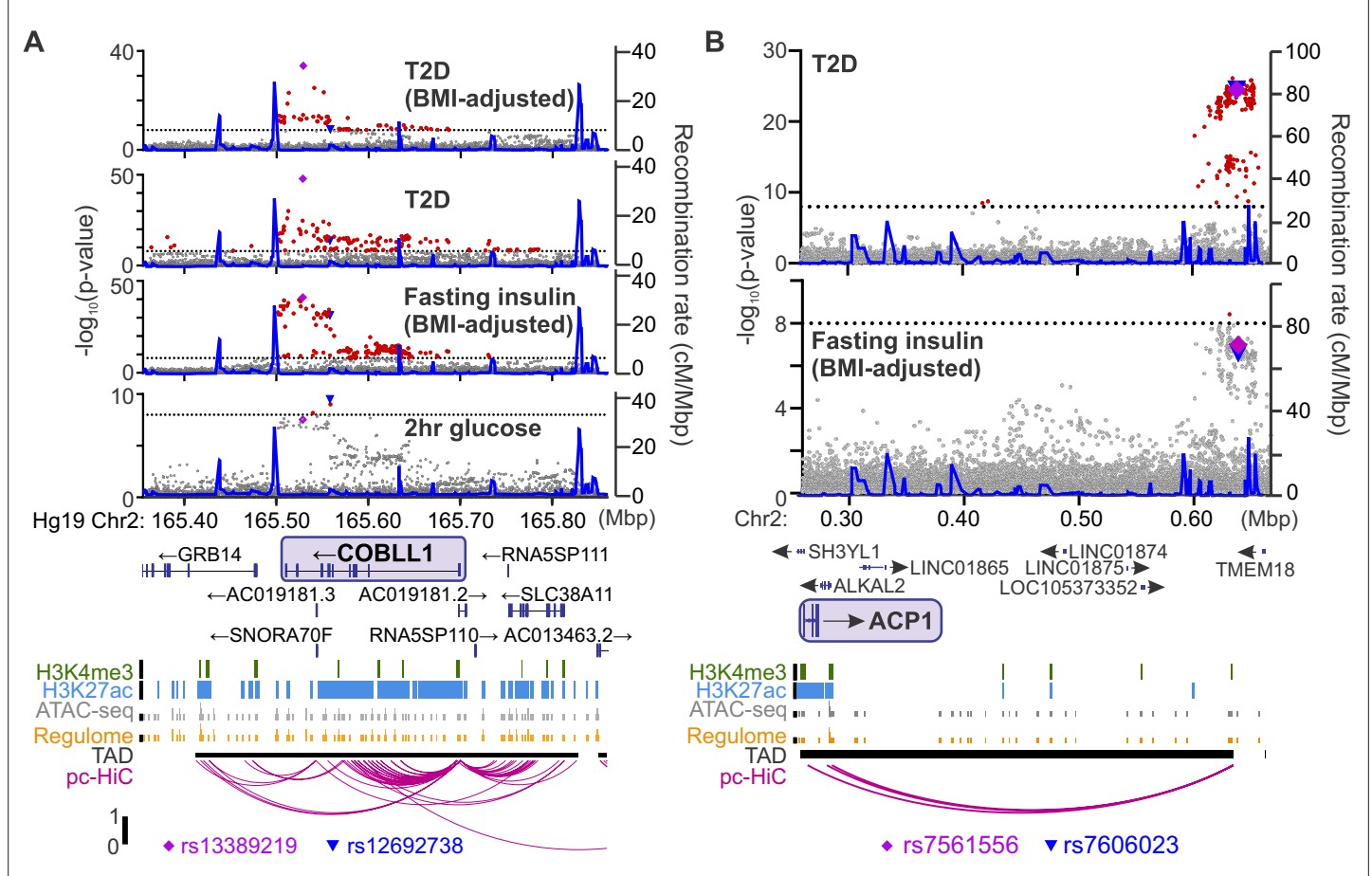

**Figure 6.** Identifying candidate protein targets by integrating human genome-wide association studies (GWAS). (**A**) An example gene, *COBLL1*, orthologous to a gene coding for a protein identified as highly correlated to Ca$^{2+}$ wave parameters in the founder mice. The recombination rate is indicated by the solid blue line. Significant single-nucleotide polymorphisms (SNPs; $8 < -\log_{10}(p)$, red) decorate the gene body for multiple glycemia-related parameters (in bold). Human islet chromatin data (***Miguel-Escalada et al., 2019***) for histone methylation (H3K4me3), histone acetylation (H3K27ac), ATAC-sequencing (ATAC-seq), and regulome score suggest active transcription of the gene within a topologically associated domain (TAD). Human islet promoter-capture HiC data (pc-HiC) (***Miguel-Escalada et al., 2019***) show contacts between the SNP-containing regions decorating the gene and its promoter. The highest SNP for 2 hr glucose (▼) and the other parameters (♦) are indicated. (**B**) Some orthologues did not show SNPs decorating the gene itself but did show looping to regions with SNPs for glycemic traits. The promoter of *ACP1*, for example, loops to a region within its topologically associated domain (black bar) with strong SNPs for type 2 diabetes risk and near-threshold SNPs for fasting insulin adjusted for body mass index (BMI). Some SNPs (▼, ♦) lie directly on the contact regions identified by HiC, whereas others lie immediately proximal to these contacts. For both panels, the significance of association ($-\log_{10}$ of the p-value) for the individual SNPs is on the left $y$ axis and the recombination rate per megabasepair (Mbp) is on the right $y$ axis. Chromosomal position in Mbp is aligned to Hg19. SNP data were provided by the Common Metabolic Diseases Knowledge Portal (https://hugeamp.org/).

region (determined using human islet promoter-capture HiC data; ***Miguel-Escalada et al., 2019***) as illustrated by ACP1 (***Figure 6B***). This yielded a list of 647 human genes strongly associated with diabetes-related SNPs. Among these genes, 478 were not previously associated with insulin secretion (see Methods), suggesting they may have understudied roles in islet function (***Figure 7A, B***; ***Supplementary file 1*** and ***Supplementary file 2***). Our approach thereby leverages the genetic diversity of the eight CC founder strains and human GWAS for diabetes-related traits to highlight genes that may play a novel role in islet function and relate to diabetes risk.

To aid in the selection of mouse strains for validating potential candidate regulators, we provide a resource with proteomic and Ca$^{2+}$ data (***Figure 7C–E***; https://data-viz.it.wisc.edu/FounderCalci-umStudy/, https://connect.doit.wisc.edu/FounderCalciumStudy/, https://doi.org/10.5061/dryad.j0zpc86jc, https://rstudio.it.wisc.edu/FounderCalciumStudy). This will enable the user to identify proteins correlated to other proteins or traits of interest, and from there, identify which strain(s) may

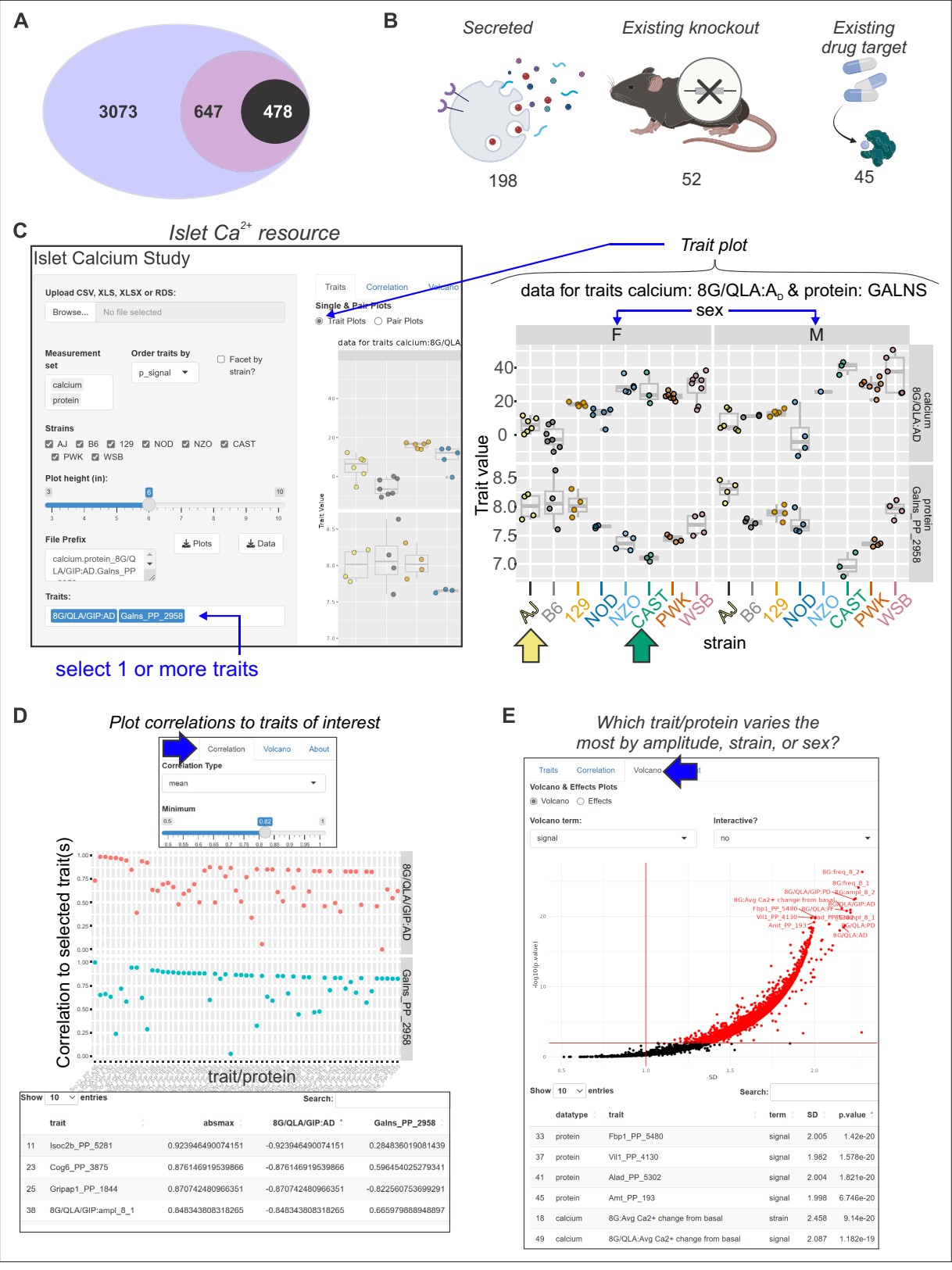

**Figure 7.** Mining Ca$^{2+}$ data using a novel online resource . (**A**) 3073 islet proteins significantly correlated to islet Ca$^{2+}$ parameters of interest. Among the proteins, 647 had orthologues containing single-nucleotide polymorphisms (SNPs) for glycemic traits. Of these, 478 showed no results in our starting triage (see Methods) under any alias, suggesting they may be understudied in islet biology. (**B**) Of these 478 proteins, 198 were found to be secreted either as soluble proteins or in exosomes (*Bateman et al., 2021*; *Thul et al., 2017*; *Uhlén et al., 2019*; *Navajas et al., 2022*; *Wang et al., 2013*; *Chen*

*Figure 7 continued on next page*

Figure 7 continued

*et al., 2019*; *Gonzales et al., 2009*), 52 have existing knockout mice with annotated glycemia or pancreatic phenotypes (*Groza et al., 2023*; *Blake et al., 2021*), and 45 have existing compounds that target them (*Stanford et al., 2021*; *Coker et al., 2019*; *Davies et al., 2015*; *Gaulton et al., 2017*; *Santos et al., 2017*; *Zhou et al., 2022*). To make these data more accessible, we have developed an online resource that enables individuals to query the $Ca^{2+}$ and proteomic data simultaneously. The user can select proteins and calcium traits (**C**) and display strain and sex distribution of these traits to determine the ideal backgrounds on which to test their traits or proteins of interest. In this example, GALNS is highly correlated to 8G/QLA/GIP $A_D$, with the highest and lowest abundance strains for GALNS being AJ (yellow arrow) and CAST (green arrow), respectively. (**D**) The user can also query for the correlations between $Ca^{2+}$ traits and proteins against one another or other traits of the same category. (**E**) The user can also see which of the traits or proteins has the largest change and most significant effects by sex, strain, or sex and strain.

be most appropriate for studies of their protein of interest. In the examples illustrated in *Figure 7C*, the user queries for the strain/sex distribution of the protein GALNS, which shows a high negative correlation to multiple traits, including the active duration time in 8G/QLA. Strains at the extremes of this trait are also extremes regarding GALNS protein abundance. Strains with high abundance (AJ, for example [yellow arrow]) would be ideal models for inhibition or knockout, whereas CAST mice (green arrow) could be a comparison strain for validating the role of the protein, as they express much less GALNS. Users can also query for the proteins or calcium parameters and see their correlation to the other proteins and/or calcium parameters (*Figure 7D*). Importantly, this includes the ability to look at the correlations between traits for individual strains or subsets of strains, enabling the user to see how the main clusters of strains (classical, wild-derived, and disease models) or individual outlier strains are drivers of specific traits. Finally, the user can see which traits or proteins show the strongest strain, sex, or sex-by-strain effects using the options in the volcano plot (*Figure 7E*). Together, these and other tools in the resource will allow researchers to explore their traits or proteins of interest as well as determine the appropriate model systems and conditions that may best interrogate their experimental questions of interest.

## Discussion

### Genetic variability drives islet function

While the development and progression of T2D is potentiated by environmental factors, an estimated 50% of disease risk is driven by genetic factors (*Dimas et al., 2014*; *Clee and Attie, 2007*; *Bergman et al., 2003*). Therefore, to study the genetic variation contributing to T2D, we took advantage of the genetic diversity contained within the eight CC founder strains. These mice collectively contain a level of genetic diversity mirroring that seen in humans, making them an excellent experimental platform to link genetics with altered islet function (*Threadgill et al., 2011*; *Svenson et al., 2012*). We demonstrate that they also vary in their $Ca^{2+}$ response to various insulin secretagogues, supporting the use of these mice to identify novel genes involved in regulating islet biology.

### Dissecting the calcium waveform highlights islet regulatory pathways

Variations in $Ca^{2+}$ dynamics are highly complex, reflecting changes in metabolism, extra-islet signaling, and $Ca^{2+}$ itself (*Merrins et al., 2022*). We therefore selected stimulatory conditions to assess each of these components in islets of the eight mouse strains. 8 mM glucose was first used to survey glycolytic responses, because we have observed that several strains reliably oscillate at this glucose concentration. Furthermore, this glucose concentration remains close to the stimulatory threshold, thus reducing the possibility of oscillations plateauing if islet $Ca^{2+}$ responses were left shifted in any strains (*Nunemaker et al., 2006a*; *Carter et al., 2009*; *Emfinger et al., 2022*). We then added QLA as fuel to engage mitochondrial metabolism and paracrine signaling from α-cells, providing a survey of α- to β-cell communication in the islet (*Lewandowski et al., 2020*; *Foster et al., 2022*; *El et al., 2021*; *Capozzi et al., 2019*). Finally, we used GIP to interrogate the islet incretin responses and the cAMP amplification pathway (*El et al., 2021*), before returning to a low glucose concentration, which enabled us to establish baseline $Ca^{2+}$ levels.

The variation in $Ca^{2+}$ response to these conditions can be better understood by examining the multitude of pathways regulating $Ca^{2+}$ dynamics. As mentioned previously, altering ionic pathways involved in regulating $Ca^{2+}$, such as $K_{ATP}$ channels, has a very different effect on $Ca^{2+}$ oscillations compared to altering glycolysis. It is important to further dissect these pathways, as changing specific components

of glucose metabolism can elicit different effects. For example, activating glucokinase (GK, which is rate limiting for glycolysis) and activating pyruvate kinase (PK, an ATP-generating enzyme which directly controls $K_{ATP}$ channel closure in the final step of glycolysis) both reduce the silent duration by accelerating $K_{ATP}$ channel closure. GK and PK activation also alter the active duration and the plateau fraction, however they do so in opposite directions (*Lewandowski et al., 2020*). Activating GK increases the active duration and oscillation period, while activating PK decreases those same parameters via $Ca^{2+}$ extrusion (*Foster et al., 2022*). The incretin hormones GLP-1 and GIP reduce silent duration and oscillation period, most likely due to their ability to activate Epac2 and sensitize $K_{ATP}$ channels to ATP-dependent closure (*Kang et al., 2008*; *Holz et al., 1993*). Thus, PK and GLP1 have a common target (i.e. $K_{ATP}$), and therefore a similar effect on silent duration. These examples illustrate the benefit of analyzing multiple $Ca^{2+}$ parameters for understanding pathways of interest.

The importance of analyzing a variety of $Ca^{2+}$ parameters is further supported by the insulin secretion measurements in the male WSB and 129 mice. While average $Ca^{2+}$ is a common metric used to predict insulin secretion, relying on only this metric would suggest that the two strains secrete insulin similarly (*Figure 3—figure supplement 1*). However, the WSB mice secreted significantly more insulin in 8G, 8G/QLA, and 8G/QLA/GIP (*Figure 3*). Based on our correlation analysis between $Ca^{2+}$ parameters and insulin secretion across each sex and strain, active duration, and pulse duration in 8G more accurately predicted insulin secretion and may be highly informative when used with other data (*Figure 4*). This is similar to results published by other groups, suggesting that average $Ca^{2+}$ does not correlate well with insulin secretion (*Heart et al., 2006*).

## Strains segregate by their phylogenetic origins

Notably, several of the strains appeared to cluster together with similar responses. One such group is composed of three classical strains (A/J, B6, and 129), which had relatively similar waveforms that were dominated by slow oscillations. These differed from a second group containing the wild-derived strains (CAST, WSB, and PWK) which closely matched one another and were dominated by faster oscillations. Additionally, even with the clear separation between the clusters, inter-strain variation was still observed within the clusters (e.g. more 129 islets had plateau responses to 8G/QLA than the B6 or AJ).

The classical strains have been highly inbred (>150 + generations) and descend from related common ancestors, the 'fancy mice'. They also have extremely low genetic diversity, with 97% of their genomes explained by fewer than 10 haplotypes (*Clee and Attie, 2007*; *Beck et al., 2000*; *Yang et al., 2011*). In contrast, wild-derived strains are each independent in their parental origin, inbred for far fewer generations than the classical strains (20 vs. >150), and include significant contributions from other subspecies of *Mus musculus* than the predominant subspecies (*M. m. domesticus*) in the classical strains, particularly CAST (*M. m. castaneous*) and PWK (*M. m. musculus*) (*Clee and Attie, 2007*; *Beck et al., 2000*; *Yang et al., 2011*). It is thus unsurprising that the two primary $Ca^{2+}$ response clusters were composed of the classical and wild-derived strains. Multiple loci have already been linked to islet dysfunction and differential metabolic homeostasis in the classic strains (*Clee and Attie, 2007*). Our work here highlights the promise in using wild-derived strains to unmask previously underappreciated islet phenomena, something we and others have previously shown (*Mitok et al., 2018*; *Lee et al., 2011*; *Kreznar et al., 2017*).

While considered one of the classical strains, the NOD mice differed from the two primary clusters noted above. They displayed a combination of features from both groups and had a high degree of inter-islet variability, especially the female mice. NOD share common ancestors with the Swiss-Webster mice, which do not share parental origin with the other classical strains (*Beck et al., 2000*) and also display a 'mixed' phenotype consisting of islet $Ca^{2+}$ oscillations in response to glucose, with both slow and fast components (*Nunemaker et al., 2006a*).

Additionally, while all NOD mice were normoglycemic, a heterogeneous response was observed in islets from female, but not male, NOD mice. Female NOD mice are known to develop islet immune cell infiltration and subsequent autoimmune diabetes, whereas males are largely protected from this (*Pearson et al., 2016*). Male NOD islets were largely consistent in their $Ca^{2+}$ waveforms. In the females, however, a high degree of heterogeneity in responses was observed across the female's islets (*Figure 1—figure supplement 2*). For any NOD female, some islets resembled those from NOD males in their clear oscillations, others largely lacked oscillatory behavior other than a strong pulse in

response to 8G/QLA, and still others had an intermediate response. These observations may reflect varying degrees of dysfunction in the NOD female islets as the mice progress to diabetes, though we cannot say whether this results from variation in β-cell intrinsic defects or islet immune cell infiltration.

The NZO mice also varied from the two clusters previously discussed. Male, and all but one of the female NZO mice were diabetic. Islets from diabetic mice had reduced amplitude and oscillatory behavior, other than a single pulse in 8G/QLA. This pattern is similar to the patterns observed in many of the NOD female islets. On the other hand, islets of the one non-diabetic female NZO mouse demonstrated clear, slow oscillations (*Figure 1—figure supplement 1B*), which was surprising given reports of low $K_{ATP}$ abundance due to *Abcc8* mutations in the NZO (*Andrikopoulos et al., 2016*) and the strong role of $K_{ATP}$ channels in regulating islet $Ca^{2+}$(*Marinelli et al., 2022b*; *Koster et al., 2000*; *Remedi and Nichols, 2016*; *Ashcroft et al., 2017*). While not of the same lineage as the NOD, the NZO do exhibit some autoimmune infiltration in the pancreas (*Junger et al., 2002*), and the marked difference between the non-diabetic and diabetic NZOs, along with the variation in female NOD islet responses, further suggests that intra-islet variability for the NOD mice may be the result of disease progression.

Understanding the genetic variation driving islet responses in the founders may be informative beyond these specific strains. Screens in the DO mice and similar outbred populations can track SNPs associated with trait variation to their parental inbred strain of origin. Our previous genetic screen for drivers of islet function observed that many of the quantitative trait loci (QTL) appearing for ex vivo islet traits had effects driven by the SNPs from the wild-derived strains as opposed to the classical inbred strains (*Keller et al., 2019*). For example, the QTL mediated by *Zfp148*, which drives $Ca^{2+}$ oscillation and insulin secretion phenotypes in β-cells (*Emfinger et al., 2022*), also had strong strain effects from wild-derived strains (*Keller et al., 2019*).

Previous studies of islet $Ca^{2+}$ have largely been confined to a handful of strains, and many studies by individual labs tend to use the strains linked with their projects. While this does include a few outbred stocks (e.g. NMRI [*Pohorec et al., 2022*; *Sterk et al., 2021*; *Stožer et al., 2021*], CD-1 [*Dahlgren et al., 2005*; *Hauke et al., 2022*; *Scarl et al., 2019*]), direct comparisons of these to traditional inbred lines are rare (*Pohorec et al., 2022*), and studies of specific genes often use traditional inbred lines, as wild-derived lines do not respond as well to the conventional assistive reproductive technologies required for genome editing and transgenesis (*Hirose et al., 2017*; *Mochida et al., 2014*).

One study, comparing the outbred NMRI stock to the C57BL/6J and C57BL/6N strains (*Pohorec et al., 2022*), found that the NMRI displayed significantly lower $Ca^{2+}$ frequencies than the C57 lines, particularly in physiological glucose ranges and had similar active periods. While highly informative, there were important differences between these studies and our studies here. Of note, the studies were done in acute slice culture, in only one sex, and the $Ca^{2+}$ frequencies detected did not resemble the (at least for the C57 lines) the slow oscillations observed in isolated islets from these inbred strains (e.g. *Figure 1*, *Lewandowski et al., 2020*; *Emfinger et al., 2022*).

## Mouse-to-human integration nominates novel islet drivers

One limitation of our current study is that the association between islet proteins and $Ca^{2+}$ waveforms is correlative and therefore cannot distinguish proteins that are causal for the differences in islet $Ca^{2+}$ between strains from proteins that change because of these differences. One approach to discriminate cause from effect, and establish the relevance to humans, is to identify whether genes encoding human orthologues of these proteins are associated with glycemic traits in human GWAS. SNPs for glycemic traits (Table 2), particularly those involving insulin, suggest that alterations in these proteins may impart disease risk, which is less likely for proteins that do not play critical regulatory roles. Thus, filtering our candidates for glycemic trait associations in human GWAS, while not definitive, suggests a likely causal role for these proteins in mediating differences in islet $Ca^{2+}$ and insulin secretion among the different mice. Integrating human GWAS data with the proteins most correlated to $Ca^{2+}$ dynamics nominated ~650 protein candidates, of which approximately one-third have been previously shown to have roles in islet biology. These include well-established drivers of insulin secretion; for example SUR1, GLUT2, and GNAS. Other previously unknown candidates show promise for validation, as they are already targets of small molecule compounds (e.g. ACP1 and others, *Stanford et al., 2021*; *Coker et al., 2019*; *Davies et al., 2015*; *Gaulton et al., 2017*; *Santos et al., 2017*; *Zhou et al., 2022*), are secreted (e.g. COBLL1 and others, *Bateman et al., 2021*; *Thul et al., 2017*; *Uhlén et al., 2015*; *Uhlén*

*et al., 2019*; *Navajas et al., 2022*; *Wang et al., 2013*; *Chen et al., 2019*; *Gonzales et al., 2009*), or have been knocked out in mice, resulting in metabolic phenotypes (*Figure 7B*, *Supplementary file 1*, and *Groza et al., 2023*; *Supplementary file 2*).

Our approach to merge human GWAS with our findings in mouse assumes that the glycemic-related SNPs we nominated alter the abundance or function of the human orthologues. Most SNPs that are strongly associated with phenotypes in human GWAS are noncoding, residing within introns, promoters, 3'UTRs, or intergenic regions (e.g. *Figure 6*). Therefore, a limitation of our approach is the assumption that SNPs regulate the gene they are proximal to, which is not always accurate (*Nyaga et al., 2018*; *Chen et al., 2020*; *Smemo et al., 2014*). To infer a more direct link between SNPs and potential target genes, we incorporated human islet chromatin data (*Miguel-Escalada et al., 2019*). Physical contact between a region containing SNPs and a distal gene supports a regulatory role, as for ACP1 (*Figure 6B*). Additionally, SNPs within regions of open chromatin (ATAC-seq) and actively transcribed regions (histone markers) suggest a higher likelihood of regulating transcription factor access. While this approach does not conclusively show a link between the SNPs and expression of the orthologue for our candidate proteins, these chromatin data more strongly suggest that the ortho-logue expression may be regulated by the candidates' SNPs.

## Exploiting strain and sex-dependent differences in $Ca^{2+}$ dynamics for model system selection

In addition to the candidate regulators with potential relevance to human islet biology, we provide a user-friendly web interface to our data where users can determine whether their gene of interest has a potential regulatory role in islets. Multiple inferences regarding the roles of specific pathways are possible via analysis of $Ca^{2+}$ oscillations in islets (*Lewandowski et al., 2020*; *Merrins et al., 2022*; *Dahlgren et al., 2005*; *Kennedy et al., 2002*), and our protein correlation data provide a resource to identify which parameter most closely correlates to a number of $Ca^{2+}$ traits. Additionally, it high-lights strain/sex outliers for a given trait or gene product, which can be used to select which strain/sex is best to explore that gene's role (e.g. *Figure 7C*). Newer technologies in reproductive assistance, transgenesis, and gene editing, together with more accurate genome sequencing and single mutations conferring docility, are quickly making utilization of the wild-derived mice more practical (*Hirose et al., 2017*; *Mochida et al., 2014*; *Karunakaran and Clee, 2018*; *Chao et al., 2019*; *Chang et al., 2017*). As many of the QTL identified in DO-based studies often have strong driver SNPs from the wild-derived strains, a further understanding of which experimental questions might be best addressed by use of these strains will be important.

We have previously provided user-friendly web interfaces that allow searches of gene expression as a function of diet (WD vs. chow, *Keller et al., 2008*; *Yau et al., 2021*) and background (e.g. BTBR and B6, *Keller et al., 2008*; *Yau et al., 2021*), correlation and QTL scans in F2 intercrosses of these mice (*Lan et al., 2006*), and where these may align with QTL in our DO studies (*Keller et al., 2019*). Many of our candidates are strongly altered by diet and have strong correlations in the F2 data for certain clinical traits including insulin and glycemic parameters (*Lan et al., 2006*). Here, we provide the correlation data for islet proteins against multiple parameters describing islet $Ca^{2+}$ responses between strains (https://doi.org/10.5061/dryad.j0zpc86jc, https://data-viz.it.wisc.edu/FounderCalci-umStudy/, https://github.com/byandell/FounderCalciumStudy, https://connect.doit.wisc.edu/Found-erCalciumStudy/, https://rstudio.it.wisc.edu/FounderCalciumStudy). These will enable researchers to better identify proteins or parameters of interest as well as appropriate background strains with which to determine the functions of these proteins.

## Materials and methods
### Chemicals

All general chemicals, amino acids, bovine serum albumin, 4-(2-Hydroxyethyl)piperazine-1-ethanesulfonic acid, N-(2-Hydroxyethyl)piperazine-N'-(2-ethanesulfonic acid) (HEPES), dimethylsulf-oxide (DMSO), glucose, glucose-dependent insulinotropic polypeptide (GIP, G2269), cOmplete Mini EDTA-free Protease Inhibitor Cocktail Tablets (11836170001), and heat-inactivated fetal bovine serum (FBS; 12306C) were purchased from Sigma-Aldrich. RPMI 1640 base medium (11-875-093), antibiotic–antimycotic solutions (15240112), NP-40 Alternative (492016), Fura Red $Ca^{2+}$ imaging dye (F3020),

**Table 1.** Imaging medium formula. Components are indicated by chemical abbreviation on the left and final concentration in mM is indicated in the right column.

| Component | Concentration (mM) |
|---|---|
| NaCl | 137 |
| KCl | 5.6 |
| MgCl$_2$ | 1.2 |
| NaH$_2$PO$_4$·H$_2$O | 0.5 |
| NaHCO$_3$ | 4.2 |
| HEPES | 10 |
| CaCl$_2$ | 2.6 |

DiR (D12731), and agarose (BP1356-500) were purchased from Thermo Fisher. Glass-bottomed culture dishes were ordered from Mattek (P35G-0-14C). Fura Red stocks were prepared at 5 mM concentrations in DMSO, aliquoted into light-shielded tubes, and stored at −20°C until day of use (5 µM final concentration). DiR was prepared in DMSO at 2 mg/ml, aliquoted to light-shielded tubes, and stored at 4°C until use. All imaging solutions were prepared in a bicarbonate/HEPES-buffered imaging medium (formula in *Table 1*). Amino acids were prepared as 100×stock in the biocarbonate/HEPES-buffered imaging medium, aliquoted into 1.5 ml tubes, and frozen at −20°C until day of use. Aliquots of GIP stock were prepared at 100 µM in water and kept at −20°C until day of use.

## Animals

Animal care and experimental protocols were approved by the University of Wisconsin-Madison Animal Care and Use Committee. These studies used the following strains: A/J (RRID:IMSR_JAX:000646), C57BL/6J (B6) (RRID:IMSR_JAX:000664), 129S1/SvlmJ (129) (RRID: IMSR_JAX:002448), NOD/ShiLtJ (NOD) (RRID:IMSR_JAX:001976), NZO/HILtJ (NZO) (RRID:IMSR_JAX:002105), CAST/EiJ (CAST) (RRID:IMSR_JAX:000928), PWK/PhJ (PWK) (RRID:IMSR_JAX:003715), and WSB/EiJ (WSB)(RRID:IMSR_JAX:001145). Most strains (B6, AJ, 129, NOD, PWK, and WSB) were bred in-house, although two strains (CAST and NZO) were purchased from Jackson Laboratory (Bar Harbor, ME). All mice were fed a high-fat, high-sucrose Western-style diet (WD, consisting of 44.6% kcal fat, 34% carbohydrate, and 17.3% protein) from Envigo Teklad (TD.08811) beginning at 4 weeks and continuing until sacrifice (aged ~19–20 weeks for all strains except the NZO males). The NZO males were sacrificed at 12 weeks of age owing to complications from severe diabetes. For each strain, three to seven males and females from at least two litters were analyzed. Animals were sacrificed by cervical dislocation prior to islet isolation.

## In vivo measurements

Fasting blood glucose and insulin levels were measured in mice at 19 weeks of age, except for the NZO males which were measured at 12 weeks of age. Glucose was analyzed by the glucose oxidase method using a commercially available kit (TR15221, Thermo Fisher Scientific), and insulin was measured by radioimmunoassay (RIA; SRI-13K, Millipore). This is the same assay that was used to measure plasma insulin for the previously published cohort used for the correlation analysis in *Figure 4*; *Mitok et al., 2018*.

## Islet imaging

Islets were isolated as previously described (*Rabaglia et al., 2005*) and incubated in recovery medium (RPMI 1640, 11.1 mM glucose, 1% antibiotic/antimycotic, 10% FBS) overnight at 37°C and 5% CO$_2$. Islets were then incubated with Fura Red (5 µM in recovery medium) at 37°C for 45 min. Imaging dishes were created from glass-bottomed 10 cm$^2$ dishes that had been filled with agarose. A channel with a central well was cut into the agarose with expanded ports on either side of the well for inflow and outflow lines. Prior to loading the chambers were perfused with the initial imaging solution (8 mM glucose in imaging medium). Islets were then loaded into these dishes. The imaging chamber was placed on a 37°C-heated microscope stage (Tokai Hit TIZ) of a Nikon A1R-Si+ confocal microscope. The solutions included 8 mM glucose (8G), 8 mM glucose + 2 mM glutamine, 0.5 mM leucine, and 1.25 mM alanine (8G/QLA), 8G/QLA + 10 nM glucose-dependent insulinotropic polypeptide (8G/QLA/GIP), and 2 mM glucose (2G), each of which were kept in a 37°C water bath. Solutions were perfused through the chamber at 0.25 ml/min for 40 min each, with constant flow controlled by a Fluigent MCFS-EZ and M-switch valve assembly (Fluigent). The scope was integrated with a Nikon

Eclipse-Ti Inverted scope and equipped with a Nikon CFI Apochromat Lambda D ×10/0.45 objective (Nikon Instruments), fluorescence spectral detector, and multiple laser lines (Nikon LU-NV laser unit; 405, 440, 488, 514, 561, and 640 nm). Bound dye was excited with the 405 nm laser and the spectral detector's variable filter was set to 620–690 nm. The free dye was excited with the 488 nm laser and the variable filter collected from 640 to 690 nm. Images were collected at 1 frame/s at 6-s intervals. Each islet was considered a region of interest for further analysis. ROI intensity was collected by NIS Elements and exported for further analysis. All microscopy was performed at the University of Wisconsin-Madison Biochemistry Optical Core, which was established with support from the University of Wisconsin-Madison Department of Biochemistry Endowment.

### Islet perifusion

Isolated islets were kept in RPMI-based medium (see above) overnight prior to perifusion, which was performed as previously described, with minor modifications (*Emfinger et al., 2022*; *Bhatnagar et al., 2011*). Islets were equilibrated in 2 mM glucose for 55 min, after which 100 µl fractions were collected every minute with the perifusion solutions set at a flow rate of 100 µl/min. All solutions and islet chambers were kept at 37°C. After the final fraction was collected, islet chambers were disconnected, inverted, and flushed with 2 ml of NP-40 Alternative lysis buffer containing protease inhibitors for islet insulin extraction.

### Secreted insulin assay

Insulin in each perifusion fraction and islet insulin content were determined using a custom assay, as previously described (*Mitok et al., 2018*). The primary (10R-I136a, also called D6C4) and secondary (61R-I136b-BT, also called D3E7) antibodies were from Biosynth.

### Imaging data analysis

Trace segments for each solution condition were analyzed using Matlab and R. Traces were detrended using custom R scripts and GraphPad PRISM. Custom Matlab scripts (*Foster et al., 2022*) (https://github.com/hrfoster/Merrins-Lab-Matlab-Scripts (*Foster, 2022*), also stored on Zenodo https://doi.org/10.5281/zenodo.6540721) determined oscillation peak amplitude, pulse duration, active duration (the time when $Ca^{2+}$ is above 50% peak amplitude), silent duration (the difference between period and active duration), plateau fraction (the fraction of overall time per pulse spent in the active duration), pulse period, and other parameters. Spectral density deconvolution for the trace segments to determine principal frequencies was done using R. Animal averages for the different parameters defined by Matlab and R were computed and graphed using custom R scripts. Figures were created using CorelDraw and Biorender. All R scripts and the citations for the relevant packages used to generate them are available via Dryad (https://doi.org/10.5061/dryad.j0zpc86jc).

### Correlation and *Z*-score calculations

Correlation analysis was performed using the imaging data measurements and our published islet protein abundance data, ex vivo static insulin secretion measurements, and in vivo measurements made in a separate cohort of mice on the WD from the same strains and sexes used in these studies (*Mitok et al., 2018*). For each imaging parameter or previously published measurement, the *Z*-score was calculated using the formula $z = (x − µ)/σ$ where $z$ is the *Z*-score, $x$ is the animal average for that trait given the strain and sex, $µ$ is the average of all animals' values for that trait, and $σ$ is the standard deviation for all animals' values for that trait. *Z*-scores were computed in R and excel for the imaging parameters and the previously published (*Mitok et al., 2018*) islet proteomic, ex vivo secretion, and in vivo measurements.

Correlation coefficients between the *Z*-score values of the imaging parameters and *Z*-scores of the previously published protein abundance, islet secretion, and in vivo traits were computed in Excel using the CORREL function. The equation used for this function is:

$$Correl(X, Y) = \frac{\sum (x − \dot{x})(y − \dot{y})}{\sum (x − \dot{x})^2 * \sum (y − \dot{y})^2}$$

**Table 2.** Categories included in single-nucleotide polymorphism (SNP) queries.
These terms were considered as glycemia related and are categorized as such on the Common Metabolic Diseases Knowledge portal, which was queried for the relevant SNPs. Also included but not listed here were variations of these terms that were adjusted for body mass index (BMI).

| Fasting hormones | Glucose related | Tolerance test | Diabetes risk |
|---|---|---|---|
| Insulin | Fasting glucose | 2 hr glucose | T1D |
| Proinsulin | Random glucose | 2 hr insulin | T2D |
| C-peptide | Hba1c | 2 hr C-peptide | |
| Fasting glc–BMI interaction | | Acute insulin response | |
| Fasting ins–BMI interaction | | SI-adjusted acute ins. Resp. | |
| Gestational diabetes/altered fast glucose in pregnancy | | AUC insulin | |
| | | AUC insulin/AUC glucose | |
| | | Corrected insulin response | |
| | | HOMA-B | |
| | | HOMA-IR | |
| | | Ins. Secretion rate | |
| | | Ins. Sensitivity | |
| | | Incremental ins. @ 30 min OGTT | |
| | | Insulin @ 30 min OGTT | |
| | | Peak ins. response | |
| | | Peak ins. Response adj SI | |

where $X$ and $Y$ are the $Z$-scores for the correlated traits/parameters, $\dot{x}$ is the population average for trait $X$ and $\hat{y}$ is the population average for trait $Y$. Traits were considered highly correlated if absolute value for their $Z$-score correlation coefficients was ≥0.5.

## Gene enrichment and human GWAS analysis

Proteins highly correlated or anticorrelated to imaging parameters were further analyzed using pathway enrichment and presence of human GWAS SNPs. Briefly, for a given parameter, pathway analysis for the highly correlated or anticorrelated proteins to that parameter was done using Enrichr (*Chen et al., 2013*; *Kuleshov et al., 2016*). Enrichr links for the subsets of proteins highly correlated to specific calcium parameters are provided in *Supplementary file 3*, which is stored on Dryad (https://doi.org/10.5061/dryad.j0zpc86jc).

For GWAS analysis, human orthologues for genes encoding the previously measured islet proteins were identified using BioMart (*Smedley et al., 2009*). For highly correlated proteins, the protein was deemed of human interest if its orthologue had SNPs for glycemia-related traits (see *Table 2*) either along the gene body, within ±100 kbp of the gene start or end, or if any region in the gene body was connected to regions with SNPs by chromatin looping. SNPs were queried using Lunaris tool of the Common Metabolic Diseases Knowledge Portal (https://hugeamp.org/). Chromatin loop anchor points for the relevant gene orthologues were identified using previously published human islet promoter-capture HiC data (*Miguel-Escalada et al., 2019*) and the alignment between these anchor loops and orthologues of interest was done using R scripts.

For those proteins having orthologues with SNPs via this analysis, we conducted further literature searches using Pubmed, Google Scholar, ChEMBL (*Davies et al., 2015*; *Gaulton et al., 2017*; *Jupp et al., 2014*), canSAR (*Coker et al., 2019*), Uniprot (*Bateman et al., 2021*), Tabula Muris (*Tabula Muris Consortium et al., 2018*), the Human Protein Atlas (*Thul et al., 2017*; *Uhlén et al., 2015*), and other resources (*Uhlén et al., 2019*; *Varshney et al., 2017*; *Lawlor et al., 2017*) to determine tissue

expression and identify any prior roles in islet biology. Figures for the relevant protein examples were created using GraphPad Prism, CorelDraw, and the WashU Epigenome Browser (*Li et al., 2019*).

We further narrowed this list by searching each of the genes and their aliases in the PubMed, Google Scholar, and Google Search Engines along with 'insulin secretion'. This allowed us to identify which genes have a known role in altering the insulin secretory pathway, and which genes may be understudied (*Figure 7A*).

## Web resource

A web resource was created to explore the islet calcium and proteomic data and their relationships (https://data-viz.it.wisc.edu/FounderCalciumStudy, https://connect.doit.wisc.edu/FounderCalcium-Study/, https://rstudio.it.wisc.edu/FounderCalciumStudy). This resource sits on an RStudio/Connect server (see https://posit.co/). It enables the user to select traits from the calcium and protein datasets to plot by strain, sex, and calcium parameters. Distinct mice were assayed for **calcium** and **protein**. Individual strains can be selected on the main menu using the checkboxes, or all strains (default) can be viewed.

The different datasets available in the main menu are:

1. **calcium**: calcium parameters and spectral density data, with stimulatory secretion conditions
2. **protein**: islet proteomic measurements
3. **basal**: average calcium in 2 mM glucose

The **calcium** data have three stimulatory conditions (8G, 8G/QLA, and 8G/QLA/GIP), which are displayed together for each calcium parameter. The proteomic data (**protein**) are displayed for each identified peptide. In rare cases of multiple peptides per gene, both gene symbol and peptide identifier (PP number) are included (e.g. Pkm_PP_1521 for the M1 isoform of the protein PKM). Desired proteins can be selected simultaneously with desired calcium parameters for correlation analysis and paired display by both datasets. The **basal** elements retained from the **calcium** data include the Average Calcium measurement for 2 mM glucose. Proteomic data were $\log_{10}$-transformed. All traits were transformed into normal scores, keeping the sample mean and variance the same.

Scatter plots display data across sex and calcium conditions. When plotting calcium against protein or basal traits, means by strain and sex are used, as the two experiments used different mice. Correlation of selected traits with all other traits in the resource use Pearson correlation on pairwise-complete data. The user can order traits by their significance or by their correlation to other selected traits.

Statistical modeling terms include strain, sex, and the strain:sex interaction, plus additional terms for comparing calcium condition with respect to strain and sex. Users can view volcano plots displaying deviation of term effects, measured as the standard deviation (SD = square root of mean square error) divided by the raw SD for that trait, against their significance (p-value after adjusting for all other model terms, presented on $-\log_{10}$ scale). In addition to the terms, a composite 'signal' captures the combined effect of terms strain:condition + strain:condition:sex using a general *F* test computation.

All data handling and web app construction for the resource were performed using R scripts in publicly available GitHub repositories, with specifics for the calcium study at https://github.com/byandell/FounderCalciumStudy (copy archived at *Yandell, 2023b*) and the general purpose analysis and web deployment package at https://github.com/byandell/foundr (copy archived at *Yandell, 2023a*).

## Statistics

For the islet perifusion insulin measurements, statistics were determined in GraphPad Prism. Fractional secretion area-under-the-curve (AUC) was determined using Prism and differences in AUCs analyzed using post-tests following two-way analysis of variance for the indicated trace segments. Islet total insulins between strains were compared using a two-tailed Student's *t*-test with Welch's correction. For *Figure 4—figure supplement 1*, the graphs, Pearson's *R*, and regression lines were created in Prism. All data analysis used individual animal averages of the islet measurements (biological replicates). Experimental numbers were determined using prior data analyses (*Keller et al., 2019*; *Mitok et al., 2018*; *Emfinger et al., 2022*). Owing to poor islet yield as a complication of their severe diabetes, in the imaging experiments the islets from some NZO mice had to be pooled with each pool considered a biological replicate.

## Acknowledgements

Funding sources. This work was supported by NIH R01DK101573, R01DK102948, and RC2DK125961 to A Attie; NIH R01DK113103, NIH R01DK127637, and VA I01B005113 to M Merrins, ADA 7-21-PDF-157 to C Emfinger, and by the University of Wisconsin–Madison, Department of Biochemistry and Office of the Vice Chancellor for Research and Graduate Education with funding from the Wisconsin Alumni Research Foundation to M Keller. Portions of the figures were created with Biorender. License files are PC24P29AVO, UA24P29B1E, TK24P29B44, QM24P29H2P, WF24P29HB3, OM24P2BPIL, and BN24P2C01S. We would also like to thank Dr. Michael Lloyd at The Jackson Laboratory for his help with setting up the Lunaris queries.

## Additional information

### Funding

| Funder | Grant reference number | Author |
|---|---|---|
| American Diabetes Association | 7-21-PDF-157 | Christopher H Emfinger |
| National Institutes of Health | R01DK101573 | Alan D Attie |
| National Institutes of Health | R01DK102948 | Alan D Attie |
| National Institutes of Health | RC2DK125961 | Alan D Attie |
| National Institutes of Health | R01DK113103 | Matthew J Merrins |
| National Institutes of Health | R01DK127637 | Matthew J Merrins |
| U.S. Department of Veterans Affairs | I01B005113 | Matthew J Merrins |

The funders had no role in study design, data collection, and interpretation, or the decision to submit the work for publication.

### Author contributions

Christopher H Emfinger, Conceptualization, Data curation, Software, Formal analysis, Funding acquisition, Validation, Investigation, Visualization, Methodology, Writing – original draft, Writing – review and editing; Lauren E Clark, Conceptualization, Data curation, Formal analysis, Validation, Investigation, Visualization, Methodology, Writing – original draft, Writing – review and editing; Brian Yandell, Resources, Data curation, Software, Formal analysis, Visualization, Writing – original draft, Writing – review and editing; Kathryn L Schueler, Shane P Simonett, Donnie S Stapleton, Kelly A Mitok, Investigation, Methodology; Matthew J Merrins, Conceptualization, Funding acquisition, Methodology, Writing – review and editing; Mark P Keller, Conceptualization, Data curation, Formal analysis, Supervision, Funding acquisition, Visualization, Methodology, Project administration, Writing – review and editing; Alan D Attie, Conceptualization, Supervision, Funding acquisition, Visualization, Methodology, Project administration, Writing – review and editing

### Author ORCIDs

Christopher H Emfinger (ID) https://orcid.org/0000-0002-9130-4194
Lauren E Clark (ID) http://orcid.org/0000-0002-0209-9716
Shane P Simonett (ID) https://orcid.org/0000-0002-9359-7808
Kelly A Mitok (ID) http://orcid.org/0000-0002-0167-3990
Matthew J Merrins (ID) http://orcid.org/0000-0003-1599-9227
Mark P Keller (ID) http://orcid.org/0000-0002-7405-5552
Alan D Attie (ID) http://orcid.org/0000-0002-0568-2261

## Ethics

We performed these studies in accordance with the recommendations within the Guide for the Care and Use of Laboratory Animals of the National Institutes of Health and all procedures and protocols were approved by the University of Wisconsin-Madison Institutional Animal Care and Use Committee (animal protocol number A005821-R01). Every effort was made to minimize suffering.

Reviewer #1 (Public Review): https://doi.org/10.7554/eLife.88189.3.sa1
Reviewer #2 (Public Review): https://doi.org/10.7554/eLife.88189.3.sa2
Author Response https://doi.org/10.7554/eLife.88189.3.sa3

---

## Additional files

### Supplementary files

• Supplementary file 1. Proteins correlated with $Ca^{2+}$ parameters that have glycemic-related single-nucleotide polymorphisms (SNPs). This includes protein IDs, gene names, gene IDs, and human orthologues for each of the proteins that correlate to one of the following metrics and have a glycemic-related SNP (see *Table 2*): basal $Ca^{2+}$, 8G $S_D$, 8G/QLA $S_D$, 8G/QLA/GIP $S_D$, 8G $A_D$, 8G $P_D$, and 8G 1st freq.

• Supplementary file 2. Proteins understudied in islet biology. This table of proteins indicates the subset of proteins meeting our selection criteria (*Supplementary file 1*) that did not have any results in Pubmed, Google Scholar, or Google for any alias and the term 'insulin secretion', suggesting that they may be understudied in islet biology. The gene symbols for the mouse gene and human orthologue are indicated. The 'Mouse?' column indicates whether a knockout mouse with metabolic phenotypes exists (identified from sources in the subsequent 'Source' column). The 'Drug?' column similarly indicates whether any source (indicated in the following 'Source' column) shows existing compound(s) targeting the protein. Finally, the 'Secreted?' column indicates whether any source (indicated in the subsequent 'Source' column) shows an isoform of the protein to be secreted.

• Supplementary file 3. Enrichments for the highly correlated and anticorrelated proteins. The Enrichr tool (*Chen et al., 2013*; *Kuleshov et al., 2016*) queries multiple databases for information regarding gene lists and queries can be stored for access later using hyperlinks. This Excel file contains five tabs. The 'Key' tab indicates the contents of the file. The 'Uniprot_IDs_correlated' and 'Uniprot_IDs_anticorrelated' tabs each, respectively, contain in their columns lists of the Uniprot IDs for peptides correlated (coefficient >0.5) or anticorrelated (coefficient <−0.5) to specific $Ca^{2+}$ parameters listed in the row labeled 'Traits'. These Uniprot IDs were queried to determine the gene names. The 'Enrichr_correlated' and 'Enrichr_anticorrelated' tabs each, respectively, contain these corresponding gene names for those proteins correlated (coefficient >0.5) or anticorrelated (coefficient <−0.5) to specific $Ca^{2+}$ parameters listed in the row labeled 'Traits'. In the row 'Enrichr Link' contains the Enrichr query hyperlinks for each protein list. For example, the Enrichr_correlated column B has the link https://maayanlab.cloud/Enrichr/enrich?dataset=affcd1912271cd603ec6e26304ac789e which is the Enrichr database search for the proteins listed in that column that correlate with the 1st frequency component in 8G. This table is available via the Zenodo repository (DOI 10.5281/zenodo.7776230).

• MDAR checklist

### Data availability

Raw data are available via Dryad (https://datadryad.org/stash/dataset/doi:10.5061/dryad.j0zpc86jc). R scripts for analysis are available through the related database Zenodo (https://doi.org/10.5281/zenodo.7776210), and Matlab scripts are also available there (https://doi.org/10.5281/zenodo.7776210) as well as on GitHub (https://github.com/hrfoster/Merrins-Lab-Matlab-Scripts). Processed data and supplemental information are on Zenodo (https://doi.org/10.5281/zenodo.7776230). The code for the web resource is available via Github (https://github.com/byandell/FounderCalciumStudy, copy archived at *Yandell, 2023b*) and the general deployment and analysis package is also there (https://github.com/byandell/foundr copy archived at *Yandell, 2023a*).

The following datasets were generated:

| Author(s) | Year | Dataset title | Dataset URL | Database and Identifier |
|---|---|---|---|---|
| Emfinger CH, Clark LE, Yandell B, Schueler KL, Simonett SP, Stapleton DS, Mitok KA, Merrins MJ, Keller MP, Attie AD | 2023 | Genetic variation in mouse islet Ca2+ oscillations reveals novel regulators of islet function | https://datadryad.org/stash/dataset/doi:10.5061/dryad.j0zpc86jc | Dryad Digital Repository, 10.5061/dryad.j0zpc86jc |
| Emfinger CH, Clark LE, Yandell B, Schueler KL, Simonett SP, Stapleton DS, Mitok KA, Merrins MJ, Keller MP, Attie AD | 2023 | Genetic variation in mouse islet Ca2+ oscillations reveals novel regulators of islet function | https://zenodo.org/record/7776210 | Zenodo, 10.5281/zenodo.7776210 |
| Emfinger CH, Clark LE, Yandell B, Schueler KL, Simonett SP, Stapleton DS, Mitok KA, Merrins MJ, Keller MP, Attie AD | 2023 | Genetic variation in mouse islet Ca2+ oscillations reveals novel regulators of islet function | https://zenodo.org/record/7776230 | Zenodo, 10.5281/zenodo.7776230 |

The following previously published datasets were used:

| Author(s) | Year | Dataset title | Dataset URL | Database and Identifier |
|---|---|---|---|---|
| Mitok KA, Freiberger EC, Schueler KL, Rabaglia ME, Stapleton DS, Kwiecien NW, Malec PA, Hebert AS, Broman AT, Kennedy RT, Keller MP, Coon JJ, Attie AD | 2018 | Islet proteomics reveals genetic variation in dopamine production resulting in altered insulin secretion | http://www.coonlabdata.com/founder_mice/main.php | Joshua Coon Laboratory Founder Mouse Studies Resource, founder_mice |
| Miguel-Escalada I, Bonàs-Guarch S, Cebola I, Ponsa-Cobas J, Mendieta-Esteban J, Atla G, Javierre B, Rolando D, Farabella I, Morgan C, García-Hurtado J, Beucher A, Morán I, Pasquali L, Ramos-Rodríguez M, Appel E, Linneberg A, Gjesing A, Witte D, Pedersen O, Grarup N, Ravassard P, Torrents D, Mercader J, Piemonti L, Berney T, de Koning E, Kerr-Conte J, Pattou F, Fedko I, Groop L, Prokopenko I, Hansen T, Marti-Renom M, Fraser P, Ferrer J | 2019 | Human pancreatic islet 3D chromatin architecture provides insights into the genetics of type 2 diabetes | https://www.crg.eu/en/programmes-groups/ferrer-lab#datasets | Centre for Genomic Regulation, ferrer-lab#datasets |

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
