## [Editor Report · eLife assessment]

The authors provide a **fundamental** resource, detailing genetic variation of nutrient-responsive islet calcium regulation in mice through the lens of proteomics. The evidence for the mechanisms identified using this resource is **compelling** and strongly supported by integration with results from genome-wide association studies in humans. The construction of a streamlined and searchable web interface for the data will maximize their accessibility and utilization by the community.

---

## [Referee Report · Reviewer #1 (Public Review)]

This paper looks at nutrient-responsive Ca++ flux in islet cells of eight genetically diverse mouse strains. The investigators correlate Ca++ flux with insulin secretory capacity, demonstrating that calcium parameters in response to different nutrients are a better predictor of insulin secretory capacity than average calcium. They also correlate Ca++ flux with previously collected islet protein abundance followed by integration with human genome-wide association studies. This integration allows them to identify a sub-set of proteins that are both relevant to human islet function and that may play a causal role in regulating islet Ca++ oscillations. All data have been deposited in a searchable public database. There are many strengths to this paper. To my knowledge, this is the first work to assess the genetics of nutrient-responsive Ca++ flux in islets. Given the importance of Ca++ for beta cell insulin secretion, this work is of high importance. Investigators also use the founders of two powerful genetic mouse models: the diversity outbred and collaborative cross, opening up several avenues of future research into the genetics of Ca++ flux. By looking at multiple parameters of Ca++ flux, investigators are able to start to understand which parameters may be driving low or high insulin secretion. Integration with protein abundance and human GWAS has allowed identification of proteins with known roles in insulin secretory capacity, as well as several novel regulators, again opening up several avenues of future research. Finally, the public database is likely to be useful to multiple investigators interested in following up specific protein targets or in conducting future genetic studies.

---

## [Referee Report · Reviewer #2 (Public Review)]

This is an interesting paper from a reputable group in the field of islet physiology. The authors have provided the results from extensive studies, which will contribute to the knowledge of islet dysfunction and diabetes pathophysiology. The authors studied "the human orthologues of the correlated mouse proteins that are proximal to the glycemia-associated SNPs in human GWAS". This implies two assumptions - (1) human and mouse proteins do not differ in terms of islet physiology and calcium signaling; (2) the proteins proximal to the SNPs are the causal factors for functional differences, though the SNPs could affect protein/gene function distant from the SNPs.

---

## [Author Response]

The following is the authors’ response to the original reviews.

We greatly appreciate the thoughtful suggestions made by the Reviewers. We have addressed all of their comments below, with our responses bulleted and in italics. We believe these changes have helped clarify the manuscript and strengthen it overall.

**Reviewer 1**
1. Figures 1B and Supp. Figure 1A: It would be worth mentioning that the wave-form in the 129 strain in response to QLA starts out like AJ and B6, but transitions to looking like the wild-derived strain. So, although not quite as drastic as the NZO and NOD strains, it is not quite like the other classical inbred strains.

• We thank the reviewer for pointing this out. We have added further language to clarify the point:

“Additionally, even with the clear separation between the clusters, inter-strain variation was still observed within the clusters (e.g. more 129 islets had plateau responses to 8G/QLA than the B6 or AJ).”

1. The figures are generally excellent and really help to clarify the work in the paper. For Figure 2A, it would help even further if you could number the six different Ca++ parameters that are measured. They're all there, but it takes a bit of time to find them on the figure and numbering will make it easier on your reader.

• We appreciate this suggestion and have implemented it in our revised Figure 2A. The Ca2+ parameters are now numbered, and the description of this figure has been adjusted accordingly in the results section.

We added the revised text in the results section:

“To elucidate strain differences in Ca2+ dynamics, we focused on six parameters of the Ca2+ waveform (Figure 2A): (1) peak Ca2+ (the top of each oscillation); (2) period (the length of time between two peaks); (3) active duration (the length of time for each Ca2+ oscillation measured at half of the peak height, also known the oxidative “secretory” phase, or “MitoOx” (8); (4) pulse duration (active duration plus extra time for Ca2+ extrusion); (5) silent duration (the electrically-silent “triggering” phase, also known as “MitoCat” (8), which culminates in KATP closure and membrane depolarization); and (6) plateau fraction (the active duration divided by the period, or the fraction of time spent in the active “secretory” phase).”

1. Figure 4A, B: I was expecting to see Ca++ vs insulin parameters in the different strains/sexes. In addition to the heat maps, it would be useful to see the regression plots, showing where each strain and sex falls for the insulin and Ca++ parameters.

• This is an excellent suggestion, and we have added a new Supplemental Figure 5 to provide examples of various strain/sex patterns that drive the correlations used for the heatmap and histogram in Figure 4A and B.

We added text in the results section referring to this point:

“Clustering the Ca2+ responses into distinct groups based on our observations of the waveforms (Figure 1B, Figure 4C-E, and Supplemental Figures 1 and 2) also occurs when correlating individual Ca2+ parameters to ex vivo secretion and clinical data (Supplemental Figure 5). For example, the anticorrelation between the 1st frequency component in 8G and percent insulin secreted in 8.3G/QLA (Supplemental Figure 5A) separates the classic inbred, wild-derived, and diabetes-susceptible strains into distinct groups despite the variability in the trait. Correlation between the silent duration in 8G/QLA to insulin secretion in 8.3G/QLA, likewise groups by strain (Supplemental Figure 5B). Finally, some correlations, such as that between 8G/QLA/GIP silent duration and plasma insulin at sacrifice (Supplemental Figure 5C), can be strongly influenced by outlier strains; e.g., NZO. Collectively, these data demonstrate that genetics has a profound influence on key parameters of islet Ca2+ oscillations.”

1. Please include methods for the insulin measurements collected in Fig. 4.

• Thank you for pointing out this missing information. We have clarified that prior insulin measurements (plasma insulin and ex vivo static insulin secretion that were used in Figure 4 for correlation analysis) were completed in another previously published cohort of mice (reference 17: Mitok KA, Freiberger EC, Schueler KL, Rabaglia ME, Stapleton DS, Kwiecien NW, et al. Islet proteomics reveals genetic variation in dopamine production resulting in altered insulin secretion. The Journal of biological chemistry. 2018;293(16):5860-77).

We added this new text (highlighted) to the results section to help clarify this point:

“Fasting blood glucose and insulin levels were measured in mice at 19 weeks of age, except for the NZO males which were measured at 12 weeks of age. Glucose was analyzed by the glucose oxidase method using a commercially available kit (TR15221, Thermo Fisher Scientific), and insulin was measured by radioimmunoassay (RIA; SRI13K, Millipore). This is the same assay that was used to measure plasma insulin for the previously published cohort used for the correlation analysis in Figure 4 (17).”

1. In the methods, please include details on the four conditions used for Ca++ imaging of the islets, and the timing for each condition.

• We appreciate this guidance in clarifying our manuscript, and we have now included the conditions and timing for each condition in the methods section.

We added the following text to the results section to help clarify this:

“The solutions included 8 mM glucose (8G), 8 mM glucose + 2 mM glutamine, 0.5 mM leucine, and 1.5 mM alanine (8G/QLA), 8G/QLA + 10 nM glucose-dependent insulinotropic polypeptide (8G/QLA/GIP), and 2 mM glucose (2G), each of which were kept in a 37°C water bath.”

**Reviewer 2**
One major critique is that the authors studied "the human orthologues of the correlated mouse proteins that are proximal to the glycemia-associated SNPs in human GWAS". This implies two assumptions - (1) human and mouse proteins do not differ in terms of islet physiology and calcium signaling; (2) the proteins proximal to the SNPs are the causal factors for functional differences, though the SNPs could affect protein/gene function distant from the SNPs.

• Thank you very much for highlighting this limitation in our study. We think this is very important to address which we have done in our discussion section.

We have added the following text to discuss this important issue:

“Our approach to merge human GWAS with our findings in mouse assumes that the glycemic-related SNPs we nominated alter the abundance or function of the human orthologues. Most SNPs that are strongly associated with phenotypes in human GWAS are noncoding, residing within introns, promoters, 3’UTRs, or intergenic regions (e.g. Figure 6). Therefore, a limitation of our approach is the assumption that SNPs regulate the gene they are proximal to, which is not always accurate (76-78). To infer a more direct link between SNPs and potential target genes, we incorporated human islet chromatin data (37). Physical contact between a region containing SNPs and a distal gene supports a regulatory role, as for ACP1 (Figure 6B). Additionally, SNPs within regions of open chromatin (ATAC-seq) and actively transcribed regions (histone markers) suggest a higher likelihood of regulating transcription factor access. While this approach does not conclusively show a link between the SNPs and expression of the orthologue for our candidate proteins, these chromatin data more strongly suggest that the orthologue expression may be regulated by the candidates’ SNPs.”